



# High contributions of fossil sources to more volatile organic carbon

Haiyan Ni[1,2,3], Ru-Jin Huang[2*], Junji Cao[2], Wenting Dai[2], Jiamao Zhou[2], Haoyue Deng[1], Anita Aerts-Bijma[1], Harro A. J. Meijer[1], Ulrike Dusek[1]

[1]Centre for Isotope Research (CIO), Energy and Sustainability Research Institute Groningen (ESRIG), University of Groningen, Groningen, 9747 AG, the Netherlands
[2]State Key Laboratory of Loess and Quaternary Geology, Key Laboratory of Aerosol Chemistry and Physics, Institute of Earth Environment, Chinese Academy of Sciences, Xi'an, 710061, China
[3]University of Chinese Academy of Sciences, Beijing, 100049, China

*Correspondence to*: Ru-Jin Huang (rujin.huang@ieecas.cn)

**Abstract.** Sources of particulate organic carbon (OC) with different volatility have rarely been investigated despite the significant importance for better understanding of the atmospheric processes of organic aerosols. In this study we develop a radiocarbon ($^{14}C$) based approach for source apportionment of more volatile OC (mvOC) and apply to ambient aerosol samples collected in winter in six Chinese megacities. mvOC is isolated by desorbing organic carbon from the filter samples in He at 200 °C in a custom-made aerosol combustion system for $^{14}C$ analysis. Evaluation of this new isolation method shows that the isolated mvOC amount agrees very well with the OC1 fraction (also desorbed at 200 °C in He) measured by a thermal optical analyzer using the EUSAAR_2 protocol. The mvOC, OC and elemental carbon (EC) of thirteen combined PM$_{2.5}$ samples in six Chinese cities are analyzed for $^{14}C$ to investigate their sources and formation mechanisms. The relative contribution of fossil sources to mvOC is 59 ± 11 %, consistently larger than the contribution to OC (48 ± 16 %) and smaller than that to EC (73 ± 9 %), despite large differences in fossil contributions in different cities. The average difference in the fossil fractions between mvOC and OC is 13 % (7 %–25 %; range), similar to that between mvOC and EC (13 %; 4 %–25 %). SOC concentrations and sources are modelled based on the $^{14}C$-apportioned OC and EC, and compared with concentrations and sources of mvOC. SOC concentrations (15.4 ± 9.0 μg m$^{-3}$) are consistently higher than those of mvOC (3.3 ± 2.2 μg m$^{-3}$), indicating that only a fraction of SOC is accounted for by the more volatile carbon fraction desorbed at 200 °C. The fossil fraction in SOC is 43 % (10 %–70 %), lower than that in mvOC (59 %; 45 %–78 %). Correlation between mvOC and SOC from non-fossil sources (mvOC$_{nf}$ vs. SOC$_{nf}$) and from fossil sources (mvOC$_{fossil}$ vs. SOC$_{fossil}$) are examined to further explore sources and formation processes of mvOC and SOC.



## 1 Introduction

Aerosol particles are of importance for atmospheric chemistry and physics, and exert a crucial effect on the climate system, air quality and human health (Fuzzi et al., 2015). Carbonaceous aerosols, consisting of organic carbon (OC) and elemental carbon (EC), comprise a large fraction of the fine aerosol mass (20 %–80 %) (Cao et al., 2007; Tao et al., 2017). OC contains
a large variety of organic species, such as polycyclic aromatic hydrocarbons and other components with potential mutagenic and carcinogenic effects, while EC contains chars, amorphous-like carbon consisting of randomly oriented poly-aromatic layers and carbon nanospheres formed from incomplete combusted carbon-based fuels (Pöschl, 2005). It should be noted that since no clear distinction between OC and EC exists, OC and EC are operationally defined based on the measurement techniques (Petzold et al., 2013). When a thermo-optical method is used to separate and determine OC and EC, EC is
described as a thermally refractory carbon continuum and OC is weakly refractory (Pöschl, 2003, 2005; Petzold et al., 2013).

EC is exclusively emitted as primary aerosols from incomplete combustion of biomass (e.g., wood, crop residues, and grass) and fossil fuels (e.g., coal, gasoline, and diesel). OC can be of both primary and secondary origin. Primary OC (POC) is emitted directly from non-fossil (e.g., biomass burning, biogenic emissions, and cooking) and fossil sources. Secondary OC (SOC) is formed in the atmosphere via atmospheric oxidation of volatile organic compounds (VOCs) (Jacobson et al., 2000;
Hallquist et al., 2009; Bond et al., 2013). High concentrations of aerosol particles have been observed in many cities, especially in China, where carbonaceous aerosols dominate particle mass concentrations, with SOC responsible for a large fraction of OC (Guo et al., 2014; Huang et al., 2014; Fang et al., 2017). Thus, understanding the sources, formation and transformation of EC, OC and SOC is important for better understanding of air pollution.

For aerosol source apportionment, radiocarbon ($^{14}$C) analysis of OC and EC has been used to quantify their fossil and non-
fossil fractions, based on the fact that emissions from fossil sources are $^{14}$C-free due to long-time decay, whereas non-fossil emissions contain the contemporary $^{14}$C content (e.g., Szidat et al., 2004, 2006; Dusek et al., 2013, 2017). Earlier $^{14}$C measurements of OC and EC found that fossil sources contribute more to EC than OC (Heal, 2014 and references therein). However, direct $^{14}$C measurements on SOC are not possible yet, due to the technical and conceptual difficulties in isolating the SOC fraction from filter samples. In positive matrix factorization (PMF) based source apportionment of aerosol mass
spectrometer (Jayne et al., 2000) dataset, oxidized organic aerosols (OOA, also referred to as secondary organic aerosol, SOA) can be separated into semi-volatile OOA (SV-OOA) and low-volatility OOA (LV-OOA) based on their volatility (e.g., Huffman et al., 2009; Wang et al., 2017). Volatility is an important physical property as it determines the partitioning between gas and particulate phases of organic species (Donahue et al., 2006, 2009). Direct measurement of OA volatility is challenging because OA is a mixture of thousands of individual organics spanning a wide range of volatilities (Donahue et
al., 2011). Different approaches have been used to estimate the OA volatility. For example, using thermodenuder (TD) coupled with an aerosol mass spectrometer (Burtscher et al., 2001; Wehner et al., 2002), the volatility of different OA components (such as hydrocarbon-like OA (HOA), biomass burning OA (BBOA) or OOA) can be estimated for ambient



aerosols. Source and ambient studies indicate that BBOA and HOA are generally more volatile than OOA. Meanwhile, the volatility of BBOA can be quite variable, depending on type of biomass and the combustion conditions, and either higher or lower than that of HOA (e.g., Grieshop et al. 2009c; Huffman et al. 2009; Paciga et al. 2016; Cao et al., 2018).

In some recent studies, a thermal desorption approach has been used for analysis of filter samples at lower and higher
temperatures as an indicator of volatility (Holzinger et al., 2013; Lopez-Hilfiker et al., 2014, 2015; Timkovsky et al., 2015; Masalaite et al., 2017, 2018). Even though the desorption temperature is not a direct measure of the particle volatility, OC desorbed from the filter at lower temperature tends to be more volatile, whereas the less volatile (i.e., more refractory) OC tends to be desorbed at higher temperatures. Vodička et al. (2015) and Masalaite et al. (2017) found that OC in urban samples was desorbed at lower temperatures compared to costal, forest, and rural background samples in Eastern Europe.
Both studies suggest that OC is more volatile in urban area close to the emission sources. Keller and Burtscher (2017) found that after aging of biomass burning emissions less OC desorbed at lower temperatures. The reduction in volatility after aging agrees with previous TD aerosol mass spectrometer studies (Grieshop et al., 2009b). Even though the thermal desorption and TD measurements are different methods, results of both methods are usually in qualitative agreement and show similar trends. Earlier studies (e.g., Grieshop et al., 2009c; Chen et al., 2007) show similar trends qualitatively. Ma et al. (2016)
made first efforts to link the OC fractions desorbed at different temperatures from filter samples on a thermal-optical analyzer to the volatility basis set, showing that OC fractions desorbed at lower temperatures (e.g., 140 °C, 280 °C) are semi-volatile.

Both POC and SOC can contribute to the OC fraction desorbed at lower temperatures. A number of previous studies have found that at least a part of POC from various sources is semi-volatile, including wood burning, gasoline and diesel vehicle
exhausts and cooking (Lipsky and Robinson, 2006; Shrivastava et al., 2006; Robinson et al., 2007; Grieshop et al., 2009b, 2009c; May et al., 2013a, 2013b). Moreover, a significant part of freshly formed SOC is semi-volatile and will contribute to the more volatile OC fraction desorbed at lower temperatures (Holzinger et al., 2010; Salo et al., 2011; Meusinger et al., 2017; Gkatzelis et al., 2018). The relative contributions from fossil and non-fossil sources to OC can be dependent on the volatility. For example, recent studies in wintertime of Lithuania have shown that vehicular sources are associated with OC
fractions desorbed at lower temperatures while biomass burning (as a non-fossil source) contributes more strongly to less volatile OC fractions desorbed at higher temperatures (Masalaite et al., 2017, 2018).

Although an increasing number of studies has shown that OC from different emission sources and/or at different aging status may have different volatility, a quantitative study of the sources of OC with different volatility is scarce. In this study, wintertime $PM_{2.5}$ samples from 6 Chinese megacities are studied. Sources of carbonaceous aerosols including different
carbon fractions such as OC, EC and SOC are estimated by radiocarbon ($^{14}$C) source apportionment. In addition, we operationally define a more volatile OC fraction (mvOC) as the carbon fraction desorbed from the filter samples at 200 °C in





helium (He), and investigate sources of mvOC. To our best knowledge, this is the first time that the source contribution to the more volatile fraction of OC is measured unambiguously by $^{14}$C.

## 2 Methods

### 2.1 Sampling

Simultaneous sampling was made during winter 2013/2014 in two northern (Beijing and Taiyuan) and three southern (Chongqing, Guangzhou and Shanghai) Chinese cities (Fig. S1). The samples from Xi'an in northern China were collected during winter 2015/2016. These sites were selected to represent urban-scale concentration and located in the university or research center campus, >100 m from local sources such as main roadways (Table S1). The 24 h integrated PM$_{2.5}$ samples were collected from 10:00 to 10:00 the next day (local standard time, LST). Filter samplers were deployed on roof-tops

about 6 to 20 m above ground level. In Xi'an, Beijing, Guangzhou and Shanghai, PM$_{2.5}$ samples were collected on pre-backed (780 °C for 3 h) quartz fiber filter (QM/A, Whatman Inc., Clifton, NJ, USA, 20.3 cm × 25.4 cm) using a high-volume sampler (TE-6070 MFC, Tisch Inc., Cleveland, OH, USA) at a flow rate of 1.0 m$^3$ min$^{-1}$. In Taiyuan and Chongqing, samples were collected on pre-baked 47-mm Whatman quartz microfiber filters (QM/A) using mini-volume samplers at a flow rate of 5 L min$^{-1}$ (Airmetrics, Oregon, USA). After sampling, all filters were packed in a pre-baked aluminum foils

(450 °C for 3 h), sealed in polyethylene bags and stored in a freezer at -18 °C until analysis.

### 2.2 Determination of carbon fractions by thermal-optical analysis

For the PM$_{2.5}$ samples collected in Xi'an, filter pieces of 1.5 cm$^2$ were taken for OC and EC analysis using a carbon analyzer (Model 5L, Sunset Laboratory, Inc., Portland, OR, USA) following the thermal-optical transmittance protocol EUSAAR_2 (Cavalli et al., 2010). The EUSAAR_2 protocol defines OC1 as the carbon fraction that desorbs in helium (He) at 200 °C for

2 min. OC1 is compared with more volatile OC (mvOC) extracted also in He at 200 °C but for 5 min using our aerosol combustion system (ACS) (Dusek et al., 2014). No charred OC is observed by the transmittance signal in the OC1 stage during the thermo-optical analysis. For PM$_{2.5}$ samples collected in Beijing, Taiyuan, Chongqing, Guangzhou and Shanghai, filter pieces of 0.5 cm$^2$ were used to measure OC and EC using a Desert Research Institute (DRI) Model 2001 Thermal/Optical Carbon Analyzer (Atmoslytic Inc., Calabasas, CA, USA) following the IMPROVE_A thermal/optical

reflectance (TOR) protocol (Chow et al., 2007). OC fractions of EUSAAR_2 protocol are desorbed in He at 200 °C (OC1), 300 °C (OC2), 450 °C (OC3), and 650 °C (OC4) in He. Different from EUSAAR_2 protocol, IMPROVE_A protocol defines OC fractions at stepwise temperature of 140 °C, 280 °C, 480 °C, and 580 °C in He. Details of the carbon fraction measurements were described in our previous work (Cao et al., 2013; Zenker et al., 2017).



### 2.3 Radiocarbon ($^{14}$C) measurements of mvOC, OC and EC

#### 2.3.1 Sample selection for $^{14}$C analysis

For $^{14}$C analysis, we selected the samples carefully to cover periods of low and high TC (= OC + EC) and PM$_{2.5}$ concentrations to get samples representative of various pollution conditions that did occur in each city. Two or three

composite samples from each city representing high (H), medium (M) and low (L) TC concentrations are selected for $^{14}$C analysis of EC, OC and mvOC (Fig. S2). Each composite sample consists of 2 to 4 24 h filter pieces with similar TC loadings and air mass backward trajectories (Fig. S2, Table S1).

#### 2.3.2 Extraction of mvOC, OC, EC

Three separate extractions were performed for mvOC, OC and EC on our aerosol combustion system (ACS) (Dusek et al.,

2014). The ACS consists of a combustion tube and a purification line. Aerosol filter pieces placed on the filter boat are heated at different temperatures in pure He or O$_2$ and oxidized through the platinum catalyst in the combustion tube. The resulting CO$_2$ is isolated and separated from other gases (e.g., NO$_x$, water vapor) in the purification line. The purification line is equipped with an oven filled with copper grains and silver wire heated at 650 °C to remove NO$_x$ and liberated halogen, a U-type tube cooled with a dry ice-ethanol mixture for water removal and a flask containing phosphorous pentoxide (P$_2$O$_5$)

for removal of any trace water. The amount of purified CO$_2$ is manometrically quantified and subsequently stored in flame-sealed glass ampoules.

mvOC is desorbed by heating the filter pieces at 200 °C in He for 5 min. After introducing the filter, the ovens are flushed with He for 10 min. Pilot tests show that the flushing time (10 min, 15 min or 60 min) before heating the filter pieces does not affect the desorbed amount of mvOC (Table S2). OC is combusted by heating filter pieces at 375 °C in pure O$_2$ for 10

min. EC is extracted from a separate filter piece after removing OC completely. First water-soluble OC is removed from the filter through water extraction (Dusek et al., 2014) to minimize the charring of organic materials (Yu et al., 2002; Zhang et al., 2012). Then, by heating the filter pieces at 375 °C in O$_2$ for 10 minutes, most water-insoluble OC can be removed. Subsequently, the oven temperature is increased to 450 °C for 3 min to remove the most refractory OC that left on the filter. However, during this step some less refractory EC might be lost. Finally, the remaining EC is combusted by heating at

650 °C in O$_2$ for 5 min (Dusek et al., 2014, 2017; Zenker et al., 2017).

Contamination introduced by the isolation procedure is determined by following exactly the same isolating procedures with either empty filter boat or with pre-heated filters (at 650 °C in O$_2$ for 10 min). The contamination introduced by the combustion process yields 0.72 ± 0.44 µgC EC, 0.85 ± 0.49 µgC OC, 0.52 ± 0.31 µgC mvOC per extraction, respectively. Compared with our sample size of 30–391 µgC EC, 30–445 µgC OC, and 15–121 µgC mvOC, the blanks are ≤7 % of the

sample amount and therefore relatively small compared to our sample sizes.



Two standards with known $^{14}C/^{12}C$ ratios are combusted using the ACS as quality control for the combustion process: an oxalic acid standard and anthracite. Small amounts of solid standard materials are directly put on the filter boat of the combustion tube and heated in $O_2$ at 650 °C for 10 min. In the further $^{14}C$ analysis, the $CO_2$ derived from combustion of the standards is treated exactly like the samples. Therefore, the contamination introduced by the combustion process can be

estimated from the deviation of measured values from the nominal values of the standards (Table S3) (Dusek et al., 2014). The contamination inferred in this indirect way is below 1.5 µgC per extraction, which is slightly higher than the directly measured contamination of OC and EC separately but in the range of a TC contamination. It is also relatively small compared to the size of OC (30–445 µgC) and EC samples (30–391 µgC) in this study.

**2.3.3 $^{14}C$ measurements by accelerator mass spectrometer (AMS)**

AMS measurements were conducted at the Centre for Isotope Research (CIO) at the University of Groningen. The extracted $CO_2$ is released from the glass ampules and directed to a gas inlet system (Ruff et al., 2007), where the sample is diluted with He to 5 % $CO_2$ (Salazar et al., 2015). The $CO_2/H_2$ mixture is directly fed into the Cs sputter ion sources of the Mini Carbon Dating System (MICADAS) AMS at a constant rate (Synal et al., 2007).

The $^{14}C/^{12}C$ ratio of an aerosol sample is usually reported relative to an oxalic acid standard (OXII) and expressed as fraction
modern ($F^{14}C$). The $^{14}C/^{12}C$ ratio of the standard is related to the unperturbed atmosphere in the reference year of 1950 by multiplying it with a factor of 0.7459 (Mook and Van Der Plicht, 1999; Reimer et al., 2004):

$$F^{14}C = \frac{(^{14}C/^{12}C)_{sample,[-25]}}{0.7459 \times (^{14}C/^{12}C)_{OXII,[-25]}}, \qquad (1)$$

where the $^{14}C/^{12}C$ ratio of the sample and standard are both corrected for machine background and normalized to $\delta^{13}C = -$
‰. Aerosol carbon from fossil sources has $F^{14}C = 0$ due to the extinction of $^{14}C$ after long-time decay. Aerosol carbon
20    from contemporary (or non-fossil) sources should have $F^{14}C \sim 1$ in an atmosphere undisturbed by human influences. However, the $F^{14}C$ values of contemporary carbon are higher than 1 due to the nuclear bomb tests that nearly doubled the $^{14}CO_2$ in the atmosphere in the 1960s and 1970s. The $^{14}CO_2$ produced by the nuclear bomb tests has been taken up by oceans and the biosphere and diluted by the $^{14}C$-free $CO_2$ emitted by the fossil fuel burning. Currently, $F^{14}C$ of the atmosphere $CO_2$ is ~1.04 (Levin et al., 2010).

25    The $F^{14}C$ values are corrected for cross contamination (also known as memory effect) (Wacker et al., 2010) using alternate measurements of HOxII and $^{14}C$-free material as gaseous standards. Correction for instrument background (Salazar et al., 2015) is done by subtracting the memory corrected $F^{14}C$ values of the $^{14}C$-free standard. Finally, the values are normalized to the average value of the (memory and background corrected) HOxII standards. All standards used for the corrections are measured on the same day as the samples.



### 2.4 Estimation of source contributions to different carbon fractions

$F^{14}C$ of EC, OC and mvOC (i.e., $F^{14}C_{(EC)}$, $F^{14}C_{(OC)}$ and $F^{14}C_{(mvOC)}$, respectively) are directly measured. We define 'more refractory organic carbon' (mrOC) as the difference between OC and mvOC. $F^{14}C$ of mrOC ($F^{14}C_{(mrOC)}$) can be calculated by isotope mass balance:

$$F^{14}C_{(mrOC)} = \frac{F^{14}C_{(OC)} \times OC - F^{14}C_{(mvOC)} \times mvOC}{OC - mvOC}. \qquad (2)$$

$F^{14}C_{(EC)}$ can be converted to the fraction of biomass burning ($f_{bb}(EC)$) by dividing with an $F^{14}C$ value representative of typical biomass burning emissions ($F^{14}C_{bb}$). Analogously, the fraction of non-fossil OC, mvOC and mrOC (i.e., $f_{nf}(OC)$, $f_{nf}(mvOC)$ and $f_{nf}(mrOC)$, respectivly) can be estimated from their corresponding $F^{14}C$ values and $F^{14}C$ of non-fossil sources ($F^{14}C_{nf}$). $F^{14}C_{bb}$ and $F^{14}C_{nf}$ are estimated as $1.10 \pm 0.05$ and $1.09 \pm 0.05$ (Lewis et al., 2004; Mohn et al., 2008; Palstra and

Meijer, 2014), respectively, based on tree-growth models and the assumption that wood burning dominates biomass burning. $F^{14}C_{bb}$ for EC is slightly bigger than $F^{14}C_{nf}$ for OC, because besides biomass burning, biogenic emissions also contribute to OC, but have a smaller $F^{14}C$ than that of biomass burning. The estimation of $F^{14}C_{bb}$ and $F^{14}C_{nf}$ has been reported in our previous study (Ni et al., 2018).

Then, $f_{bb}(EC)$ can be used to determine EC mass concentrations from non-fossil biomass ($EC_{bb}$) and fossil fuel combustion

($EC_{fossil}$):

$$EC_{bb} = EC \times f_{bb}(EC), \qquad (3)$$

$$EC_{fossil} = EC - EC_{bb}. \qquad (4)$$

Analogously, mass concentrations of OC, mvOC and mrOC from non-fossil sources ($OC_{nf}$, $mvOC_{nf}$ and $mrOC_{nf}$, respectively) and fossil sources ($OC_{fossil}$, $mvOC_{fossil}$ and $mrOC_{fossil}$, respectively) can be determined.

Secondary OC (SOC) includes SOC from fossil ($SOC_{fossil}$) and non-fossil sources ($SOC_{nf}$):

$$SOC = SOC_{fossil} + SOC_{nf}. \qquad (5)$$

Fraction fossil in total SOC ($f_{fossil}(SOC)$) can be formulated as:

$$f_{fossil}(SOC) = \frac{SOC_{fossil}}{SOC_{fossil} + SO_{nf}}. \qquad (6)$$

For the following calculations we assume that $SOC_{nf}$ can be approximated by OC from non-fossil sources excluding primary

biomass burning ($OC_{o,nf}$; OC other non-fossil). In principle, $OC_{o,nf}$ includes $SOC_{nf}$ and non-fossil primary OC from vegetative detritus, bioaerosols, resuspended soil organic matter, or cooking. But the contributions from plant detritus, bioaerosols and spores to $PM_{2.5}$ are likely small. If cooking sources are significant then this assumption results in an upper limit of $SOC_{nf}$.

$$SOC_{nf} \cong OC_{o,nf} = OC_{nf} - POC_{bb}, \qquad (7)$$





where $OC_{o.nf}$ can be calculated by the difference between $OC_{nf}$ and primary OC from biomass burning ($POC_{bb}$). $POC_{bb}$ is calculated by multiplying $EC_{bb}$ with the primary OC/EC ratio of biomass burning ($r_{bb}$):

$$POC_{bb} = EC_{bb} \times r_{bb}. \tag{8}$$

$SOC_{fossil}$ is calculated by subtracting primary fossil OC ($POC_{fossil}$) from $OC_{fossil}$. $POC_{fossil}$ is estimated by multiplying $EC_{fossil}$

with primary OC/EC ratio of fossil fuel combustion ($r_{fossil}$):

$$SOC_{fossil} = OC_{fossil} - POC_{fossil}, \tag{9}$$

$$POC_{fossil} = EC_{fossil} \times r_{fossil}. \tag{10}$$

Fossil sources in China are almost exclusively from coal combustion and vehicle emissions, thus $r_{fossil}$ can be estimated as

$$r_{fossil} = r_{coal} \times p + r_{vehicle} \times (1-p), \tag{11}$$

where $r_{coal}$ and $r_{vehicle}$ are the primary OC/EC ratio of coal combustion and vehicle emissions, respectively. The $r_{bb}$, $r_{coal}$ and $r_{vehicle}$ varies with the fuel types and properties, combustion conditions, sampling and analysis methods etc. Best estimate of $r_{bb}$ ($4 \pm 1$; average $\pm$ SD), $r_{coal}$ ($2.38 \pm 0.44$), and $r_{vehicle}$ ($0.85 \pm 0.16$) is done through a literature search and described in our earlier studies (Ni et al., 2018). $p$ is the fraction of EC from coal combustion ($EC_{coal}$) in $EC_{fossil}$.

We used two different methods to estimate $p$. (1) Since both coal combustion and vehicle emissions do not contain [14]C, they
can not be distinguished by [14]C measurements alone. Therefore, $p$ is randomly chosen from 0–1, that is no constraint on $p$ values. (2) EC from coal combustion is on average more enriched in the stable carbon isotope [13]C compared to vehicle emissions. Therefore, complementing [14]C results of EC with measurements of the [13]C/[12]C ratios of EC (expressed as $\delta^{13}C_{EC}$ in Eq. S1; Supplement S1) allows separation of $EC_{fossil}$ into $EC_{coal}$ and EC from vehicle emissions ($EC_{vehicle}$). Samples taken from Beijing, Shanghai and Guangzhou using high-volume samplers had enough material for analysis of both $F^{14}C_{EC}$ and
$\delta^{13}C_{EC}$, thus $EC_{coal}$, and $EC_{vehicle}$ are separated as described in detail in Supplement S2. In brief, the fraction of coal combustion and vehicle emissions in EC can be calculated from measured $F^{14}C_{EC}$ and $\delta^{13}C_{EC}$ for ambient EC combined with the source signatures. Bayesian Markov-Chain Monte Carlo (MCMC) calculations were used to account for the uncertainties in the source signatures and the measurement uncertainties (Andersson, 2011; Andersson et al., 2015). The results of the Bayesian calculations are the posterior probability density functions (PDFs) for the relative contributions of each source to
EC ($f_{coal}$ and $f_{vehicle}$; Fig. S3). The $p$ can be formulated as:

$$p = \frac{f_{coal}}{f_{fossil}} = \frac{f_{coal}}{f_{coal} + f_{vehicle}} \tag{12}$$

The PDF of $p$ is derived from the PDF of $f_{coal}$ and $f_{vehicle}$, and shown in Fig. S4.

To propagate uncertainties, a Monte Carlo simulation with 10000 individual calculations was conducted. For each individual calculation, $F^{14}C_{(EC)}$, $F^{14}C_{(OC)}$, $F^{14}C_{(mvOC)}$, concentrations of EC, OC and mvOC are randomly chosen from a normal



distribution symmetric around the measured values with the experimental uncertainties as standard deviation (SD; Table S4). For $F^{14}C_{bb}$, $F^{14}C_{nf}$, $r_{bb}$, $r_{coal}$ and $r_{vehicle}$ random values are chosen from a triangular frequency distribution with its maximum at the central value and is 0 at the lower limit and upper limit. For $p$ ranging from 0 to 1 (no $^{13}C$ constraints), $p$ is randomly chosen from a uniform distribution. For $p$ constrained by $F^{14}C$ and $\delta^{13}C$ using MCMC (hereafter $^{13}C$-constraint $p$), random

values from the respective PDF of $p$ were used (Fig. S4). In this way 10000 different estimation of $f_{bb}(EC)$, $f_{nf}(OC)$, $f_{nf}(mvOC)$, $f_{nf}(mrOC)$, $EC_{bb}$, $EC_{fossil}$, $OC_{nf}$, $OC_{fossil}$, $mvOC_{nf}$, $mvOC_{fossil}$, $mrOC_{nf}$, $mrOC_{fossil}$, $SOC_{nf}$, $SOC_{fossil}$, $SOC$ and $f_{fossil}(SOC)$ can be calculated (Tables S5, S6, S7, S8). The derived average represents the best estimate, and the SD represents the combined uncertainties.

## 3 Results

### 3.1 Method evaluation and quality control for mvOC extraction

The separation of OC and EC for $^{14}C$ analysis using our aerosol combustion system (ACS) were thoroughly evaluated by Dusek et al. (2014, 2017). It is thus necessary to validate the new extraction method for isolating mvOC. The reproducibility of the extracted mvOC amount was tested for 2 independent test filters with mvOC loadings of 6 and 18 µg cm⁻², respectively (Fig. S5). The coefficient of variation was determined as a measure of reproducibility. The reproducibility was

found to be ~5% (n = 9).

Since carbon fractions (e.g., OC1) at different desorption temperatures have mostly been measured using the EUSAAR_2 protocol in many previous studies (e.g., Vodička et al., 2015; Keller and Burtscher, 2017), our goal is to define the mvOC fraction as representative of OC1. Therefore, the mvOC is desorbed at 200 °C, the same temperature as used for OC1 in the EUSAAR_2 protocol. However, the extracted amounts on the ACS system might differ due to different heating rates and

length of the temperature step. The winter samples from Xi'an as well as the two test filters described above are used to compare mvOC concentrations from the ACS system to OC1. For most samples, excellent agreement was found between mvOC and OC1 (Fig. 1a), and most data points fall close to the 1:1 line. However, there are 3 data points deviating largely from the 1:1 line (red circle and square in Fig. 1). The two red squares represent the mvOC extraction for sample winter-H and winter-M using larger filter pieces (i.e., more mvOC in µgC per extraction). With larger filter pieces, the mvOC

(µgC/cm² filter area) extracted by the ACS system is significantly lower than that measured by the Sunset analyzer. The recoveries of these two outliers are 0.59 for sample winter-H and 0.74 for winter-M, calculated by dividing mvOC mass by OC1 mass. For winter-H, the low recovery is also repeatable with the same filter area (Fig. S5). The low recoveries for large filter pieces may result from the lower temperature (< 200 °C) towards the ends of the filter boat. At the relatively low temperature of 200 °C only a 3.5 cm long at the centre of a 12 cm combustion tube was maintained at 200 °C and outside

the 3.5 cm the temperature is lower than 200 °C, e.g., ~170 °C at the end of the combustion tube measured by a thermocouple. When filter pieces are large and placed outside the centred 3.5 cm, the desorption temperature for part of the



filter pieces will lower than 200 °C, leading to lower desorbed mvOC amount. Another possibility is saturations of the catalyst (platinum, Pt) in the combustion line of ACS system. Pt works as catalyst by collecting oxygen atoms on the surface as has been demonstrated by direct observation of an ultra-thin oxygen layer on the Pt surface by a microscope (Spronsen et al., 2017). This is used to oxidize CO and hydrocarbons to $CO_2$ in a reducing atmosphere. Thus, for large sample amounts it

is possible that the oxygen on the catalyst could not be sufficient to oxidize all desorbed CO and hydrocarbons to $CO_2$. However, we observed recoveries near 100% for mvOC amounts up to 120 μg, which was higher than the total amount desorbed for the winter-H sample. Therefore, limited catalyst capacity is not the likely explanation for the low recoveries. For subsequent experiments we consequently placed the filter pieces carefully in the 3.5 cm long 200 °C section of the combustion tube and avoid stacking multiple filter pieces to ensure a desorption temperature of 200 °C and sufficient helium

supply.

To examine the effect of the low recoveries of mvOC on the $F^{14}C_{(mvOC)}$, we compare the $F^{14}C_{(mvOC)}$ of winter-M and winter-H samples for both high and low mvOC recoveries. $F^{14}C_{(mvOC)}$ for low recoveries is roughly 0.05 (absolute value, Fig. 1b) higher than for high recoveries, which is non-negligible compared to the measurement uncertainty. In addition, to validate the the measured $F^{14}C_{(mvOC)}$ for the combined sample winter-M, we also extracted mvOC separately from those three filters

that were combined for the composite winter-M sample. Figure 2 shows the $F^{14}C_{(mvOC)}$ of combined winter-M and those of the 3 individual filter samples. From the $F^{14}C_{(mvOC)}$ and mvOC mass of the individual filters, we can estimate the expected $F^{14}C_{(mvOC)}$ of the combined winter-M sample using the isotope mass balance equation. The $F^{14}C_{(mvOC)}$ of winter-M calculated from individual filter pieces is 0.524, which is quite similar with the measured $F^{14}C_{(mvOC)}$ of 0.529 ± 0.007 for the combined winter-M with recovery close to 1.

Taken together, we conclude that $F^{14}C_{(mvOC)}$ of samples with mvOC recoveries of ~1 are reliable and used in the following discussion. The mvOC recovery of sample winter-L (red circle in Fig. 1) is also low (0.51), and we could not repeat it due to the limited filter material. For sample winter-L, we take the OC1 concentrations as mvOC and the measured $F^{14}C_{(mvOC)}$ values but assign a bigger absolute uncertainty of 0.05, due to its low mvOC recoveries. This is based on the difference in $F^{14}C_{(mvOC)}$ for winter-H and winter-M with low and high mvOC recoveries which is roughly 0.05 (Fig. 1b).

**3.2 mvOC, OC and EC concentrations**

Figure 3 shows the concentrations of mvOC, OC and EC and the mvOC contributions to OC (%) for the selected samples in the 6 Chinese megacities. mvOC and OC concentrations averaged 3.3 ± 2.2 μg m⁻³ (0.7–7.4 μg m⁻³; range) and 30.0 ± 13.8 μg m⁻³ (8.8–50.4 μg m⁻³), respectively. EC concentrations ranged from 2.5 μg m⁻³ to 14.8 μg m⁻³, with an average of 6.9 ± 3.6 μg m⁻³. High TC concentrations were found in Taiyuan (60 μg m⁻³ for sample Taiyuan-H), Chongqing (59 μg m⁻³ for

Chongqing-H), Beijing (57 μg m⁻³ for Beijing-H) and Xi'an (57 μg m⁻³ for winter-H) in descending order. Of these cities, Chongqing is located in southern China, where there is no official heating season using coal in winter. This study



nevertheless indicates severe pollution of carbonaceous aerosols in Chongqing. TC concentrations in the other southern Chinese cities (Shanghai and Guangzhou) were much lower than that in Chongqing (Fig. 3).

The fraction of mvOC in total OC (mvOC/OC in Fig. 3) gives an indication of OC volatility. The mvOC contributed on average $10.5 \pm 3.3$ % to OC, ranging from 3 % to 15 %. The mvOC/OC varies between samples within the same city and between cities, indicating complicate sources and atmospheric processing of OC. The variations might also be partially attributed to the different protocols used for OC quantification. OC in Xi'an is measured with the EUSAAR_2 protocol (up to 650 °C for desorbing OC), whereas the IMPROVE_A protocol (up to 580 °C for OC) was used for the other five cities. However, Han et al. (2016) found that the absolute OC concentrations determined by EUSAAR_2 do not differ much from those determined by IMPROVE_A ($22.6 \pm 12.0$ µg m$^{-3}$ vs. $19.7 \pm 10.7$ µg m$^{-3}$) for one-year PM$_{2.5}$ samples in Xi'an during 2012/2013. Because of the small differences of OC between the two protocols, we think the comparison of OC concentrations and mvOC/OC amongst the six cities is justified. The rest of OC (~90 %) was contributed to the mrOC. mrOC concentrations averaged $26.8 \pm 12.0$ µg m$^{-3}$, ranging from 7.9 µg m$^{-3}$ for sample Guangzhou-L to 43.1 µg m$^{-3}$ for Beijing-H.

Direct comparison of our results with previous works is somewhat difficult because different thermal/optical protocols were used. The averaged mvOC concentration ($3.3 \pm 2.2$ µg m$^{-3}$) in winter for the six studied sites in China is higher than winter concentrations at an urban background site ($1.6 \pm 1.7$ µg m$^{-3}$), a rural background site ($0.7 \pm 0.6$) in Prague (Vodička et al., 2015), where 200 °C in He was also applied to desorb this OC fraction using EUSAAR_2 protocol. The mvOC/OC ratio in our study is smaller than that of ambient samples from various other locations and much smaller than that of fresh source samples. For example, the urban background site in Prague had a mvOC/OC ratio of 28 % and the rural background site of 17 % (Vodička et al., 2015). The contribution of OC1 to total OC was as high as ~60 % for primary biomass burning measured by EUSAAR_2 thermal/optical protocol (Keller and Burtscher, 2017). For vehicle emissions, the OC fractions desorbed at 140 °C and 280 °C are the major OC fractions measured by the IMPROVE_A protocol, contributing ~30 % and ~20 % to total OC in tunnels in Taiwan (Zhu et al., 2014; 2010). Using the same protocol for OC analysis, Tian et al. (2017) found that OC fraction desorbed at 140 °C and 280 °C contributed ~13 % and ~20 % to primary OC from residential coal combustion in China. The mvOC desorbed at 200 °C, fall between 140 °C and 280 °C, thus the mvOC fraction in total OC should be higher than the faction of OC desorbed at 140 °C in total OC and lower than the fraction of OC desorbed up to 280 °C in total OC.

OC from different emission sources has different volatility, and the atmospheric processing can also alter its volatility. Keller and Burtscher (2017) found that aging reduces the volatility of OC from biomass burning, i.e., the contribution of OC1 fraction to total OC decreases from 60 % to 25 % after aging. The photochemical processing of OA can lead to accumulation of carbon in the more refractory organic fraction and also larger organic compounds (Masalaite et al., 2017). The mvOC



fraction in OC of our ambient samples is much smaller than that of the primary sources, suggesting that atmospheric aging of OC plays an important role on modifying the volatility of OC.

### 3.3 Non-fossil and fossil fractions of different carbon fractions

Figure 4a shows the fraction of non-fossil carbon in mvOC, OC and EC (respective $f_{nf}$(mvOC), $f_{nf}$(OC) and $f_{bb}$(EC)). There

are no considerable changes in $f_{nf}$(mvOC), $f_{nf}$(OC) and $f_{bb}$(EC) between polluted days ("H" samples) and clean days ("L" samples) within each site (Fig. 4a), despite the very different concentrations of carbonaceous aerosols (Fig. 3). However, $f_{nf}$(mvOC), $f_{nf}$(OC) and $f_{bb}$(EC) varied significantly among different sites: the lowest values are always found in Taiyuan, and highest in Chongqing. This implies that different pollution patterns exist in individual Chinese cities. The smallest $f_{nf}$(mvOC), $f_{nf}$(OC) and $f_{bb}$(EC) in Taiyuan suggests that fossil sources are the main contributor to mvOC, OC and EC, whereas the

largest values in Chongqing shows that the non-fossil contributions to mvOC, OC and EC are evidently higher in Chongqing than in other sites.

Ranging from 22 % to 55 %, $f_{nf}$(mvOC) (41 ± 11 %) is consistently smaller than $f_{nf}$(OC) (52 ± 16 %; 29 %–77 %) and larger than $f_{bb}$(EC) (27 ± 9 %; 10 %–42 %), despite their variations among the cities. The absolute difference in the non-fossil fractions between mvOC and OC is 13% (7 %–25 %), similar to that between mvOC and EC (13%; 4 %–25 %). Consistently

smaller $f_{nf}$(mvOC) than $f_{nf}$(OC) suggests that mvOC is more fossil (less non-fossil) than the total OC. Liu et al. (2017) also found that $F^{14}C$ of OC desorbed at lower temperature (up to 200 °C) in He was 0.389, smaller than $F^{14}C$ of total OC desorbed in the He phase of EUSAAR_2 protocol ($F^{14}C$ = 0.495, up to 650 °C) for a single test sample collected in winter in Xinjiang, China. To our best knowledge, these are the first $^{14}C$ measurements of the more volatile fraction of OC and we can unambiguously conclude that mvOC is more fossil than OC in six Chinese cites. The fraction of non-fossil carbon in mrOC

($f_{nf}$(mrOC)) is calculated by the differences between OC and mvOC using the isotope mass balance. Since mvOC is only a small fraction of OC, $f_{nf}$(mrOC) is very similar to $f_{nf}$(OC).

Primary OC from vehicular emissions is generally more volatile (i.e., less refractory) than OC from biomass burning, which is in line with the ambient results presented. For example, Grieshop et al. (2009c) constrains the volatility distribution of primary OA from a diesel engine and wood burning using measurements of TD coupled to aerosol mass spectrometer and

found that OA from wood burning is less volatile than from diesel exhaust. Chen et al. (2007) measured OC/EC from fresh biomass burning emissions and found that high-temperature OC fractions (desorbed from 450 °C to 550 °C) is the major fraction of OC, in contrast to gasoline and diesel exhausts where OC fractions desorbed at temperature lower than 250 °C are more abundant (Watson et al., 1994; Chow et al., 2004). Zhu et al. (2010, 2014) also found that OC desorbed at 140 °C is the major fraction of OC in fresh vehicle emissions in tunnel experiments. A more recent study of Masalaite et al. (2017) found

that OC desorbed at 200 °C in He has higher contributions from vehicular emissions than OC desorbed at higher temperature (250–350 °C) at three sites (urban, coastal and forest) in Lithuania, based on $\delta^{13}C$ of OC desorbed at different temperature



step. However, the conclusions of this study remained qualitative, because $\delta^{13}C$ of organic carbon can be changed significantly by isotopic fractionation during atmospheric processing, whereas the $^{14}C$ source apportionment used in our study is independent of atmospheric processing.

By comparing $(mvOC/OC)_{fossil}$ (i.e., the fraction of $mvOC_{fossil}$ in $OC_{fossil}$) and $(mvOC/OC)_{nf}$, we can get a measure of the
volatility of OC from fossil and non-fossil sources, respectively:

$$\frac{(mvOC/OC)_{fossil}}{(mvOC/OC)_{nf}} = \frac{mvOC_{fossil}/mvOC_{nf}}{OC_{fossil}/OC_{nf}} = \frac{\frac{1}{f_{nf}(mvOC)}-1}{\frac{1}{f_{nf}(OC)}-1}. \tag{13}$$

$f_{nf}(mvOC)$ is smaller than $f_{nf}(OC)$, this is equivalent to $(mvOC/OC)_{fossil} > (mvOC/OC)_{nf}$ (Eq. 13). That is, OC from fossil sources contains a larger more volatile fraction than OC from non-fossil sources. This is true for all studied cities, despite the variation of $f_{nf}(mvOC)$ and $f_{nf}(OC)$ in different cities. $(mvOC/OC)_{fossil}$ (14 ± 6.6 %; 3.7 %–28 %) is consistently higher and
more variable than $(mvOC/OC)_{nf}$ (7.5 ± 2.9 %; 2.6 %–11.6 %). $(mvOC/OC)_{fossil}$ and $(mvOC/OC)_{nf}$ in general tracks the variation of mvOC/OC, except that $(mvOC/OC)_{fossil}$ for Chongqing (28% for Chongqing-H and 21% for Chongqing-L) is much higher than the other sites (averaged 12 %). When comparing the mvOC/OC within each city, it is found that $(mvOC/OC)_{fossil}$ changes more strongly between H and L samples than $(mvOC/OC)_{nf}$, indicated by the bigger absolute differences of $(mvOC/OC)_{fossil}$ between H and L samples than that of $(mvOC/OC)_{nf}$ (Fig. 4b). Correlations between
$f_{nf}(mvOC)$ and $f_{nf}(OC)$, and between $f_{nf}(mvOC)$ and $f_{bb}(EC)$ are examined. $f_{nf}(mvOC)$ correlates more closely with $f_{nf}(OC)$ ($r^2$ = 0.89) than with $f_{bb}(EC)$ ($r^2$ = 0.62) (Fig. 5). This suggests that primary combustion sources including biomass burning and fossil fuel combustion (i.e., EC sources) can not fully explain the variation of $f_{nf}(mvOC)$, and indicates the importance of secondary formation of mvOC and/or other non-fossil contribution to mvOC besides primary biomass burning.

It is likely that some of mvOC are of secondary origin, since SOC is formed in the atmosphere and subsequently condenses
onto the aerosol particles (Hallquist et al., 2009; Jimenez et al., 2009). It has been shown that SOC formed in chambers initially desorbed at relatively low temperatures (e.g., 200 °C) (Holzinger et al., 2010; Salo et al., 2011; Meusinger et al., 2017; Gkatzelis et al., 2018). However, there is also evidence that the formation of highly oxidized low-volatile compounds can occur in SOA formation (Ehn et al., 2014). We thus compared concentrations and sources of SOC to those of mvOC. SOC concentrations and $f_{fossil}(SOC)$ are estimated based on the $^{14}C$-apportioned OC and EC in combination with $p$ values of
0–1 (Sect. 2.4). Where $\delta^{13}C$ measurements were available for high-volume filter samples (Beijing, Shanghai and Guangzhou), we compare the SOC concentrations and $f_{fossil}(SOC)$ derived from $p$ values of 0–1 with $p$ values constrained by $\delta^{13}C$. In these three cities, SOC concentrations and $f_{fossil}(SOC)$ without constraints of $p$ value (0–1) is very similar to that with constraints of $p$ value using $\delta^{13}C$, only with relatively larger uncertainties (Fig. 6). This indicates that the unconstrained $p$-values does not



lead to significant bias in SOC concentrations and $f_{fossil}$(SOC) and the main gain in using $\delta^{13}$C is currently a decrease in uncertainty.

When choosing $p$ randomly from 0 to 1, the calculated median $f_{fossil}$(SOC) is smaller than 0 for samples in Chongqing, reflecting negative SOC$_{fossil}$ values. Since SOC$_{fossil}$ is calculated by subtracting POC$_{fossil}$ from OC$_{fossil}$, this indicates POC$_{fossil}$

is overestimated in Chongqing. Because the primary OC/EC ratio is higher for coal ($r_{coal}$) than vehicle ($r_{vehicle}$) emissions, an overestimate of POC$_{fossil}$ is the direct result of overestimated $p$ values, i.e. a too high fraction of coal burning in EC$_{fossil}$ (Eq. 11). Chongqing is in southern China, where there is no official heating using coal in winter and smaller contribution of coal combustion is expected than in northern China. Further, earlier studies found that vehicle emissions contribute more to EC than coal combustion in Chongqing (Chen et al., 2017). We thus tried to lower $p$ values to 0–0.5. Lower $p$ values lead to

higher $f_{fossil}$(SOC) (Fig. 6c) and to a lesser extent higher SOC concentrations (Fig. 6b). For Chongqing, we took $p$ from 0–0.5 as a conservative estimate. For other cities except Chongqing, we kept $p$ values of 0–1 due to the very similar $f_{fossil}$(SOC) and SOC concentrations derived from $p$ values with and without constrains by $\delta^{13}$C, especially for the two southern cities in China, e.g., Shanghai and Guangzhou.

$f_{fossil}$(mvOC) ranged from 45 % to 78 %, with an average of 59 %, whereas $f_{fossil}$(SOC) ranged from 10 % to 70 %, averaged

43 %. $f_{fossil}$(mvOC) is always larger than $f_{fossil}$(SOC) (Fig. 6c), showing that mvOC is more fossil than SOC. However, both SOC and mvOC have a larger contribution of fossil sources than the total OC. The smallest differences between $f_{fossil}$(SOC) and $f_{fossil}$(mvOC) are found in Beijing for both samples Beijing-H and Beijing-L with the absolute difference of 3 % and 2 %, respectively, within the uncertainty of the measurements. Much higher differences are found in other cities, with the absolute differences ranging from 13 % to 35 %, well outside the measurement uncertainties.

Median values for SOC, SOC$_{fossil}$ and SOC$_{nf}$ are very close to their mean values (Fig. S6, Tables S7, S8). To compare with mvOC, here we use mean (± SD) as well for SOC, SOC$_{fossil}$ and SOC$_{nf}$. Lowest SOC concentrations are found to be 2.95 μg m$^{-3}$ for Guangzhou-L and highest to be 35.9 μg m$^{-3}$ for Beijing-H. SOC concentrations (15.4 ± 9.0 μg m$^{-3}$) are consistently higher than that of mvOC (3.3 ± 2.2 μg m$^{-3}$, Fig. 6a), showing that the contribution of SOC to total OC is considerably larger than that of mvOC. The major fraction of SOC has desorption temperatures above 200 °C and falls into our operationally

defined more refractory OC. SOC$_{fossil}$ ranged from 0.90 to 25.3 μg m$^{-3}$, with an average of 6.8 ± 6.4 μg m$^{-3}$, which is consistently higher than mvOC$_{fossil}$ ranging from 0.5 to 5.4 μg m$^{-3}$ with an average of 2.1 ± 1.4 μg m$^{-3}$ (Table S6). mvOC$_{nf}$ is smaller than SOC$_{nf}$ as well (Fig. S7), averaged mvOC$_{nf}$ concentrations (1.5 ± 1.1 μg m$^{-3}$) is only 1/5 of that SOC$_{nf}$ (8.5 ± 5 μg m$^{-3}$).

We examined the correlation between mvOC and SOC from non-fossil sources (mvOC$_{nf}$ vs. SOC$_{nf}$) and from fossil sources

(mvOC$_{fossil}$ vs. SOC$_{fossil}$). Comparing absolute concentrations between different cities and pollution levels is expected to result in reasonably positive correlations for all species, because it can be expected that all carbon fractions are higher in



polluted cities than in clean cities. However, it is still interesting to compare the correlation coefficients relative to one another. $mvOC_{nf}$ includes primary and secondary mvOC from non-fossil sources. Good correlation was found between $mvOC_{nf}$ and $SOC_{nf}$ ($r^2 = 0.87$, Fig. 7), which is larger than the correlation between $mvOC_{nf}$ and $POC_{bb}$ ($r^2 = 0.71$). This indicates that secondary formation of OC from non-fossil sources explains more of the variation of $mvOC_{nf}$ than primary

biomass burning. However, $SOC_{nf}$ is estimated by subtracting primary OC from biomass burning ($POC_{bb}$) from $OC_{nf}$, where $POC_{bb}$ is estimated by multiplying $EC_{bb}$ with the OC/EC ratio of fresh biomass burning plumes ($r_{bb}$) (Eq. 7). The $r^2$ for a correlation of $mvOC_{nf}$ and $SOC_{nf}$ is therefore affected by $r_{bb}$. For example, with lower $r_{bb}$ of 3 (mean), the correlation between $SOC_{nf}$ and $mvOC_{nf}$ is stronger ($r^2 = 0.92$) than for our best estimate of $r_{bb}$ (4) ($r^2 = 0.87$). For $r_{bb}$ of 5, it is slightly weaker ($r^2 = 0.76$, Fig S8), but overall $r^2$ is not very sensitive to reasonable variations in $r_{bb}$. Thus, our choice of $r_{bb}$ will not

change our conclusion that $SOC_{nf}$ correlates well with $mvOC_{nf}$, and the correlation is stronger than that between $POC_{bb}$ and $mvOC_{nf}$. $OC_{nf}$, the sum of $POC_{bb}$ and $SOC_{nf}$, is well-constrained by $F^{14}C_{OC}$ and not affected by $r_{bb}$. $mvOC_{nf}$ correlates more strongly with $OC_{nf}$ ($r^2 = 0.91$) than with $SOC_{nf}$ ($r^2 = 0.87$) or $POC_{bb}$ ($r^2 = 0.71$), suggesting strong impacts from non-fossil emissions on the variability of $mvOC_{nf}$, where the secondary formation from non-fossil VOCs explains more of the variation than primary biomass burning emissions.

$SOC_{fossil}$ is produced by the oxidation of VOCs from fossil origins, including coal combustion and vehicle emissions, whereas $mvOC_{fossil}$ can have both primary and secondary sources. No correlation is found between $mvOC_{fossil}$ and $SOC_{fossil}$ ($r^2 = 0.01$, Fig. 7c), implying that in cities or pollution conditions characterized by high concentrations of fossil SOC, $mvOC_{fossil}$ is not concurrently elevated. However, demonstrated by the good correlation between $POC_{fossil}$ and $mvOC_{fossil}$ ($r^2 = 0.69$; Fig. 7c) we can conclude that primary fossil sources may play an important role on the variation of $mvOC_{fossil}$. Larger

$f_{fossil}(mvOC)$ than $f_{fossil}(SOC)$ also indicate that fossil sources are associated more with mvOC than SOC. In Fig. 7c there are two outliner datapoint of sample Beijing-H with strongly elevated $SOC_{fossil}$ and lower $POC_{fossil}$ (shown in red circle and diamond, respectively). The highest $SOC_{fossil}/POC_{fossil}$ ratio (3.1) is found in Beijing-H, suggesting the largest fossil SOC formation during polluted days in Beijing compared to clean days in Beijing (Beijing-L with $SOC_{fossil}/POC_{fossil}$ ratio of 1.4) and samples in other cities (average $SOC_{fossil}/POC_{fossil}$ ratio of 0.7). This is also found in an early study that during winter

2013, $SOC_{fossil}/POC_{fossil}$ ratios in Beijing (with an average of 4.2) were much higher than those in Xi'an, Shanghai and Guangzhou (average $SOC_{fossil}/POC_{fossil}$ ratio of 1.3) (Zhang et al., 2015). This pollution event of Beijing-H is thus not included in the regression.

## 4 Discussion

In this study, samples are collected on a single bare-quartz (BQ) filters. mvOC may be influenced by absorption of organic

vapors to the quartz fibers (positive sampling artifact). In another word, the measured mvOC in the earlier discussion include particulate mvOC and gaseous carbon adsorbed to the quartz filters that can also be desorbed from the filters up to 200 °C.



This artifact can be estimated by using a backup quartz filter behind the BQ filter (quartz behind quartz, QBQ). The QBQ filter is assumed to be exposed to the same organic vapor concentrations as the front BQ filter and the carbon measured on the QBQ filter is mainly due to adsorbed organic vapor and can serve as an estimate of the positive artifact (Chow et al., 2010; Zhu et al., 2012).

Without the QBQ back filters in this study, we estimate the particulate fraction of mvOC ($X_{p\_mvOC}$) on the BQ filters using the OC fractions measured by the thermal/optical carbon analyzer in combination with an empirical volatility basis set (VBS) model, following the method of Ma et al. (2016) (Supplement S3). The VBS bins the organic aerosol (OA) compounds according to the effective saturation concentrations ($C^*$). For example, if $C^*$ is equal to the concentrations of organic aerosols ($C_{OA}$), then 50 % of OA mass will be in the particle phase (Donahue et al., 2006).  Ma et al. (2016) provide an

empirical $C^*$ values for each desorption temperature corresponding to the measured particulate fraction on the filter, determined by the comparison of BQ and QBQ filters. They show that these empirical $C^*$ values ($10^{1.6}$ for $OC_{140°C}$, $10^{1.1}$ for $OC_{280°C}$, $10^{0.6}$ for $OC_{480°C}$, $10^{0.1}$ for $OC_{580°C}$, where the subscript refers to the desorption temperature of the IMPROVE_A protocol) are useful to predict particulate fractions for various OC loadings and sampling environments. Assuming OA/OC mass ratio is of 1.8 for urban environment (Xing et al., 2013), the particulate fraction of the collected OC for each of the four

IMPROVE_A temperature steps $i$ $(X_{pi})$ can be estimated as a function of OC concentration, as shown in Fig. 8a. The particulate fraction of OC increases considerably from $OC_{140°C}$ to $OC_{580°C}$ and with increasing OC concentration. The desorption temperature for mvOC is 200 °C, falling within the 140 °C for $OC_{140°C}$ and 280 °C for $OC_{280°C}$. We thus estimated the $X_{p\_mvOC}$ will fall within the $X_{p1}$ for $OC_{140°C}$ and $X_{p2}$ for $OC_{280°C}$. Particulate fraction of mvOC is estimated by interpolating the particulate fraction of $OC_{140°C}$, $OC_{280°C}$ (Fig. 8a). $X_{p\_mvOC}$ ranged from 0.54 for sample winter-L with the lowest OC

concentration of 15 μg m$^{-3}$ to 0.79 for sample Beijing-H with the highest OC concentration of 50 μg m$^{-3}$ (Fig. 8b). However, for our study we consider this is a lower estimate of particulate fraction of mvOC (i.e., higher limits of the positive artifact):

i.    Our sampling duration (24 h) is much longer than that of Ma et al (2016) which is 45 min or 1 h.  With longer sampling duration, the loading on the quartz filter of both particulate OC and adsorbed organic vapors increase initially, however, the latter will reach a saturation point and level out. Thus, the percent bias by the adsorbed

organics is smaller for longer sampling duration (Turpin et al., 2000).  For example, Turpin et al. (1994) found that the loading of adsorbed organics reached to around 1.7 μg/ cm$^2$ and remained constant after 200 min of sampling time, but the aerosol loading increased linearly with the sampling time.

ii.    The empirically determined $C^*$ is affected by the sampling conditions, for example, temperature. The $C_i^*$ are defined at 300 K, higher than the average temperature in winter in China. The lower temperature drives the semi-

volatile OC into the particle phase and leads to higher particulate fraction of each OC fraction (Donahue et al., 2006).

iii.    The adsorption of organic vapors is also influenced by the face velocity (the volumetric flow rate divided by the exposed filter area). As the face velocity decreases, the adsorption of organic gaseous species increases (McDow



and Huntzicker, 1990; Turpin et al., 2000; Viana et al., 2006). In this study, most of samples were collected using high-volume samplers (HiVol) with a flow rate of 1 m$^3$ min$^{-1}$, including samples collected in Xi'an, Beijing, Guangzhou and Shanghai. Only samples in Taiyuan and Chongqing were collected using mini-volume samplers (MiniVol) at a flow rate of 5 L min$^{-1}$. The face velocities for HiVol and MiniVol sampler were 42 cm s$^{-1}$ and 7 cm s$^{-1}$, respectively. Turpin et al. (1994) found that the apparent OC concentrations on the front quartz filter were 22% greater with a face velocity of 20 cm s$^{-1}$ than those with a face velocity of 40 cm s$^{-1}$. A more recent study of Kim et al. (2016) found the positive artifacts for OC using sampler with the face velocity of 42 cm s$^{-1}$ is 7.2 %, which is smaller than 12.2% using sampler of a face velocity of 24 cm s$^{-1}$ in spring and fall of 2014 in Korea. We do not take this into consideration and apply the same $C_i^*$ for HiVol and MiniVol samples, due to the lake of $C_i^*$ for different face velocities.

Due to the lack of information on how sampling duration, temperature, and face velocity affect the $C^*$ for each carbon fraction, we treat the particulate fraction in Fig. 8b as a rough estimate. But with long sampling duration, low temperature and high face velocities for most of samples as well as high OC concentrations in winter in this study, the particulate fraction of mvOC on the collected filters should be considerably higher than the estimate derived from VBS. For example, Cheng and He (2015) found that the positive artifacts to OC was lowest (~10 %) in winter, much lower than that in summer (~23 %) in Beijing, China. With the same face velocity of ~10 cm s$^{-1}$, the smaller positive artifacts in winter than that in summer can be attributed to higher OC concentrations (35 μg m$^{-3}$ in winter vs. 12 μg m$^{-3}$ in summer) and lower temperatures in winter. We consider the positive sampling artifact in this study is small and the collected mvOC on the bare quartz filter is dominated by the particulate mvOC.

Even thought the positive artifact introduced by the adsorbed organic vapors is likely small, it is worth to investigate its potential effect on the $^{14}$C results. The measured F$^{14}$C$_{(mvOC)}$ on the BQ filter can be explained by the F$^{14}$C of particulate mvOC (F$^{14}$C$_{(p\_mvOC)}$) and adsorbed organic vapors (F$^{14}$C$_{(g\_mvOC)}$). The F$^{14}$C$_{(p\_mvOC)}$ can be calculated as:

$$F^{14}C_{(p\_mvOC)} = \frac{F^{14}C_{(mvOC)} - (1 - X_{p\_mvOC}) \times F^{14}C_{(g\_mvOC)}}{X_{p\_mvOC}}, \qquad (14)$$

where $X_{p\_mvOC}$ is the estimated particulate fraction of mvOC as shown in Fig. 8b, which is a very much lower limit as discussed earlier. $(1 - X_{p\_mvOC})$ is thus an estimate of positive artifact. How close the measured F$^{14}$C$_{(mvOC)}$ to the F$^{14}$C$_{(p\_mvOC)}$ depends on the F$^{14}$C$_{(g\_mvOC)}$.

Therefore, we conducted a sensitivity study based on the following assumption: the positive artifact is dominated by semi-volatile gases that actively partition between gas and particle phase. As the base case, we assume that F$^{14}$C$_{(g\_mvOC)}$ = F$^{14}$C$_{(mvOC)}$ and consider extreme cases of F$^{14}$C$_{(g\_mvOC)}$ = 1.5 × F$^{14}$C$_{(mvOC)}$ and F$^{14}$C$_{(g\_mvOC)}$ = 0.5 × F$^{14}$C$_{(mvOC)}$, i.e. a variation of ± 50 %. This allows F$^{14}$C values for the adsorbed gas phase as low as for EC and higher than for OC in most cases. When the





lower limit of $F^{14}C_{(g\_mvOC)}$ is used (50% lower than $F^{14}C_{(mvOC)}$), the $f_{nf}(p\_mvOC)$, calculated by dividing $F^{14}C_{(p\_mvOC)}$ with $F^{14}C_{nf}$, are from 0.04 to 0.19 (absolute value), higher than $f_{nf}(mvOC)$ (Fig. 8), but still lower than $f_{nf}(OC)$ and $f_{nf}(SOC)$ for most of samples, at least for highly-polluted (H) samples with higher particulate fraction of mvOC. That is, our conclusion that mvOC is more fossil than OC and SOC is still valid for most of samples, especially since this sensitivity study assumes

(i) an upper limit of the positive artifact as well as (ii) a very large maximum difference between the $F^{14}C$ of the adsorbed gas phase and the semi-volatile particle phase. The upper limit of $f_{nf}(p\_mvOC)$ corresponds to a semi-volatile gas phase with $F^{14}C$ values in the range of EC, which is probably not realistic.

## 5 Conclusions

Radiocarbon measurements are conducted for EC, OC and mvOC of $PM_{2.5}$ samples collected in six Chinese megacities at

wintertime. To our best knowledge, this is the first time that the sources of more volatile OC are measured unambiguously using $^{14}C$. mvOC is isolated by desorbing organics in He at 200 °C using our custom-made aerosol combustion system with reproducibility of ~5%. This new isolation method is evaluated by comparing with the OC1 fraction of the EUSAAR_2 protocol using a thermal/optical analyzer. The mvOC amount agrees very well with the OC1, despite the different heating rate and length of temperature step, showing the representativeness of the isolated mvOC. The measured $F^{14}C_{(mvOC)}$ for a

combined sample (0.529 ± 0.007), which consists of 3 individual daily filters, is quite similar to the weighted average of the individual filters. This further validates our isolation method for mvOC that the $F^{14}C_{(mvOC)}$ is representative of the combined samples. With long sampling duration, low temperature and high face velocities as well as high OC concentrations in winter in this study, the positive artifacts to mvOC should be relatively small and play a limited role on the $F^{14}C_{(mvOC)}$.

mvOC/OC ranges from 3 % to 15 %, with an average of 10.5 ± 3.3 %. The ambient samples in this study have a smaller

mvOC/OC than ambient samples from other places, and have a much smaller mvOC fraction than fresh source samples, likely due to the effects from atmospheric aging processes. mvOC/OC from fossil sources averages 14 ± 6.6 %, consistently higher and more variable than that from non-fossil sources (7.5 ± 2.9 %). That is, OC from fossil sources contains a larger more volatile fraction than that from non-fossil sources.

The non-fossil contribution to mvOC, OC and EC ($f_{nf}(mvOC)$, $f_{nf}(OC)$, $f_{bb}(EC)$) varies significantly among samples in

different sites: the lowest values are always found in Taiyuan, and the highest values in Chongqing. This implies that different pollution patterns exist in individual Chinese cities. Despite the variations of $f_{nf}(mvOC)$, $f_{nf}(OC)$, $f_{bb}(EC)$ in different cities, mvOC is more fossil than OC and less fossil than EC in all locations. The fossil contribution to mvOC, OC and EC are on average 59 ± 11 %, 48 ± 16 % and 73 ± 9 %, respectively. Higher fossil contribution to mvOC than to OC is consistent with conclusions from $\delta^{13}C$ of OC desorbed at different temperatures for ambient samples in other locations

(Masalaite et al., 2017, 2018). $f_{nf}(mvOC)$ correlates better with $f_{nf}(OC)$ ($r^2 = 0.89$) than $f_{bb}(EC)$ ($r^2 = 0.62$) (Fig. 5). This suggests that primary combustion sources including biomass burning and fossil fuel combustion (i.e., EC sources) can not





fully explain the variation of $f_{nf}$(mvOC), and indicates the importance of secondary formation of mvOC and/or other non-fossil contribution to mvOC besides primary biomass burning.

SOC concentrations and sources are modeled based on the $^{14}$C-apportioned OC and EC. SOC concentrations (15.4 ± 9.0 µg m$^{-3}$) are consistently higher than that of mvOC (3.3 ± 2.2 µg m$^{-3}$), showing that the contribution of SOC to total OC is
considerably larger than that of mvOC. A major fraction of SOC has desorption temperatures above 200 °C and falls into our operationally defined mrOC. mvOC is more fossil than SOC. The relative contribution of fossil sources to mvOC is on average 59 % (45 %–78 %; range), which is higher than that to SOC (43 %; 10 %–70 %).

Good correlation is found between mvOC$_{nf}$ and SOC$_{nf}$ ($r^2$ = 0.87, Fig. 7), which is stronger than the correlation between mvOC$_{nf}$ and POC$_{bb}$ ($r^2$ = 0.71). This indicates that secondary formation of OC from non-fossil sources has a higher
contribution than primary biomass burning to the variation of mvOC$_{nf}$. The variation of mvOC$_{fossil}$ is mainly explained by a good correlation with POC$_{fossil}$ ($r^2$ = 0.69) and no correlation is found between mvOC$_{fossil}$ and SOC$_{fossil}$ ($r^2$ = 0.01). This indicates that primary mvOC from fossil sources may play an important role in the variation of mvOC$_{fossil}$.

**Author contributions**

UD, RJH, HN, and JC conducted the study design. HN, HD, WD, and JZ carried out the experimental work.  HN, UD, AAB,
and HAJM analysed data. HN wrote the manuscript. HN, UD, and RJH interpreted data and edited the manuscript. HN prepared display items. All authors commented on and discussed the manuscript.

**Competing interests**

The authors declare that they have no conflict of interest.

**Acknowledgments**

This work was supported by the National Key Research and Development Program of China (no. 2017YFC0212701), the National Natural Science Foundation of China (NSFC; no. 91644219 and 41877408), and a KNAW project (no. 530-5CDP30). The authors acknowledge the financial support from the Gratama Foundation. Special thanks are given to Dipayan Paul, Marc Bleeker and Henk Been for their help with the AMS measurements at CIO, and to Dicky van Zonneveld for her help with $^{14}$C data correction at CIO.





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



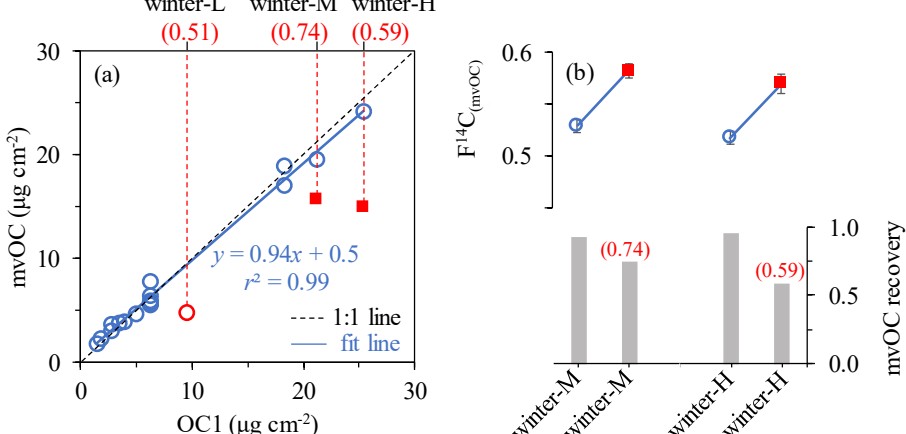

**Figure 1. (a)** OC1 loading (µg cm$^{-2}$) measured with a thermo-optical method (EUSAAR_2 protocol using a Sunset Analyzer)
and mvOC loading measured using our custom-made aerosol combustion system. The outliers are shown in red circle and
5   square with their sample name on the top $x$-axis. **(b)** F$^{14}$C$_{(mvOC)}$ of winter-M and winter-H with high and low recoveries with
respect to OC1. The red numbers in brackets in panel a and b give the respective outliers' mvOC recoveries to OC1.





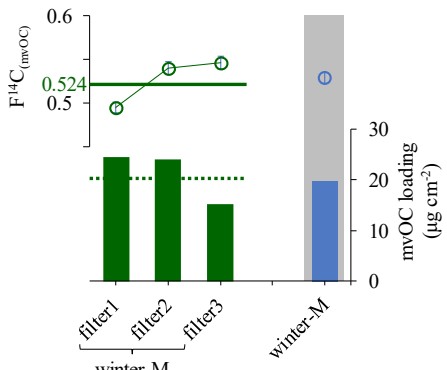

**Figure 2.** $F^{14}C_{(mvOC)}$ (shown in circle) and mvOC loading (vertical bar) of the the 3 daily filters ("filter1", "filter2" and "filter3" shown in dark green) that comprise winter-M and the composite winter-M (shown in blue in the dashed area) with

5    mvOC recovery of ~1. The dashed line in dark green indicates the expected mvOC loading and the horizontal in solid green the $F^{14}C_{(mvOC)}$ for the combined winter-M (0.524). The latter is calculated from the mvOC loading (µg cm$^{-2}$) and $F^{14}C_{(mvOC)}$ of the 3 daily filters using the isotope mass balance equation. $F^{14}C_{(mvOC)}$ uncertainties are indicated but are too small to be visible.



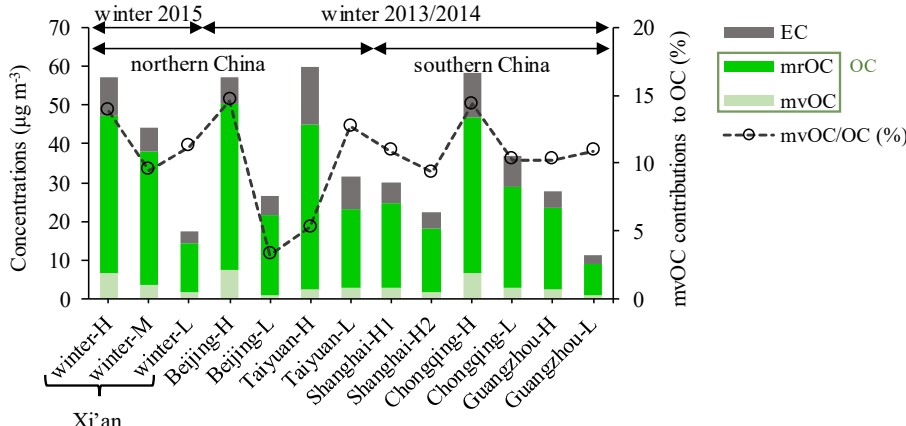

**Figure 3.** Concentrations of mvOC, mrOC and EC (µg m$^{-3}$) and contribution of mvOC to total OC (%) in 3 cities in northern

China (Xi'an, Beijing and Taiyuan) and 3 in southern China (Shanghai, Chongqing, Guangzhou). mrOC concentrations are

5   calculated by the difference between OC and mvOC.





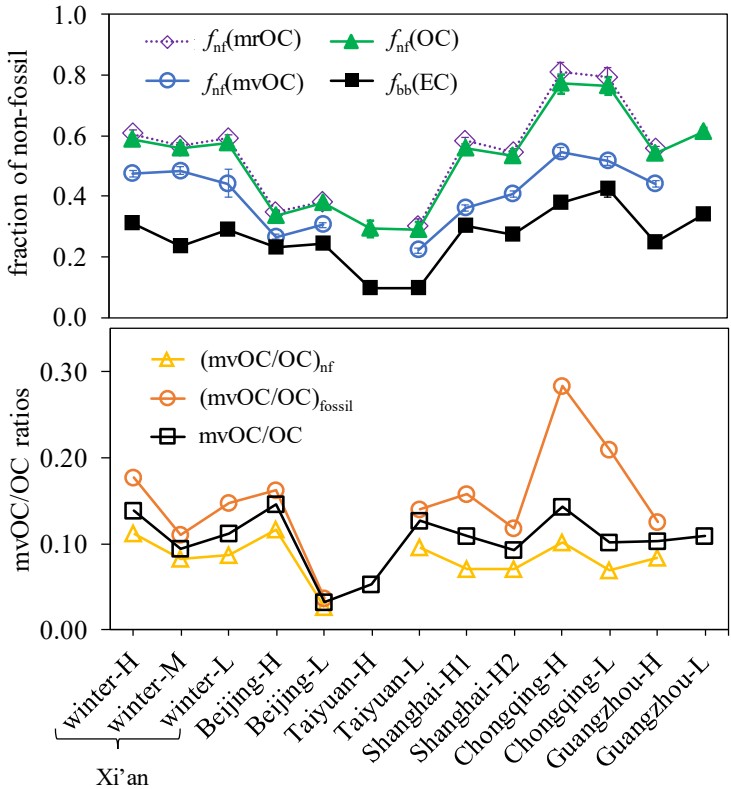

**Figure 4. (a)** Fraction of non-fossil ($f_{nf}$) in mvOC, OC and EC. $f_{nf}$(mrOC) is calculated by the differences between OC and
5  mvOC using isotope mass balance equation. **(b)** mvOC/OC ratios for non-fossil, fossil and total OC.





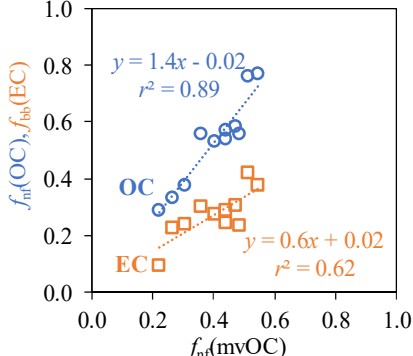

**Figure 5.** Correlation between $f_{nf}$(mvOC) and $f_{nf}$(OC), $f_{nf}$(mvOC) and $f_{bb}$(EC) for Xi'an, Beijing, Taiyuan, Shanghai, Chongqing and Guangzhou at wintertime.




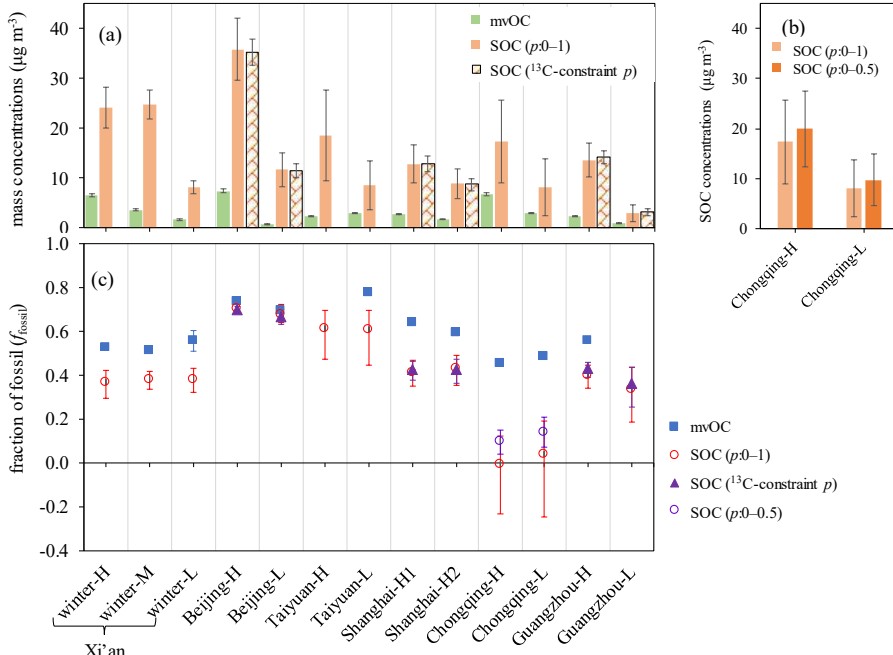

**Figure 6. (a)** Observed mvOC concentrations (show as green bar) and estimated SOC concentrations by apportioned OC and EC in combination with $p$ values without ($p$=0–1) and with constraints by $\delta^{13}$C ($^{13}$C-constraint $p$), shown as bars filled in orange and pattern fill with stripes. Error bars indicate the uncertainties of mvOC and SOC concentrations calculated by Monte Carlo error propagation. **(b)** For Chongqing, lower $p$ values of 0–0.5 are used to estimate SOC concentrations, and compared with $p$ values of 0–1. **(c)** Fraction of fossil ($f_{\text{fossil}}$) in mvOC (blue square) and SOC. $f_{\text{fossil}}$ of SOC is estimated using $p$ values without constraints ($p$=0–1) are shown as red circles and with constraints by $\delta^{13}$C ($^{13}$C-constraint $p$) as purple triangles, respectively. For Chongqing, lower $p$ values of 0–0.5 are used (purple circle). Interquartile range (25th-75th percentile) of the median $f_{\text{fossil}}$ (SOC) are shown as vertical bars, uncertainties of $f_{\text{fossil}}$(mvOC) are indicated but are too small to be visible.



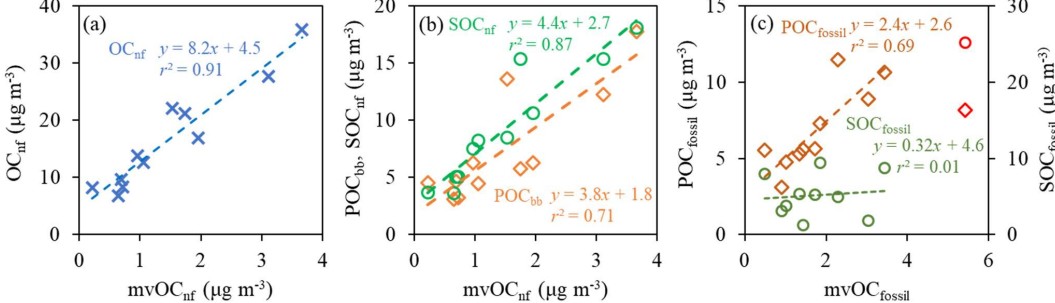

**Figure 7.** Correlation between **(a)** $OC_{nf}$ and $mvOC_{nf}$ (blue cross), **(b)** $POC_{bb}$ and $mvOC_{nf}$ (orange diamond), $SOC_{nf}$ and $mvOC_{nf}$ (green circle), **(c)** $POC_{fossil}$ and $mvOC_{fossil}$ (diamond in dark orange), $SOC_{fossil}$ and $mvOC_{fossil}$ (circle in dark green) for Xi'an, Beijing, Taiyuan, Shanghai, Chongqing and Guangzhou at wintertime. In panel c, the data points marked in red are from sample Beijing-H with the considerably higher $SOC_{fossil}/POC_{fossil}$ ratio (3.1) than other samples (an average of 0.8) and are not included in the regression.





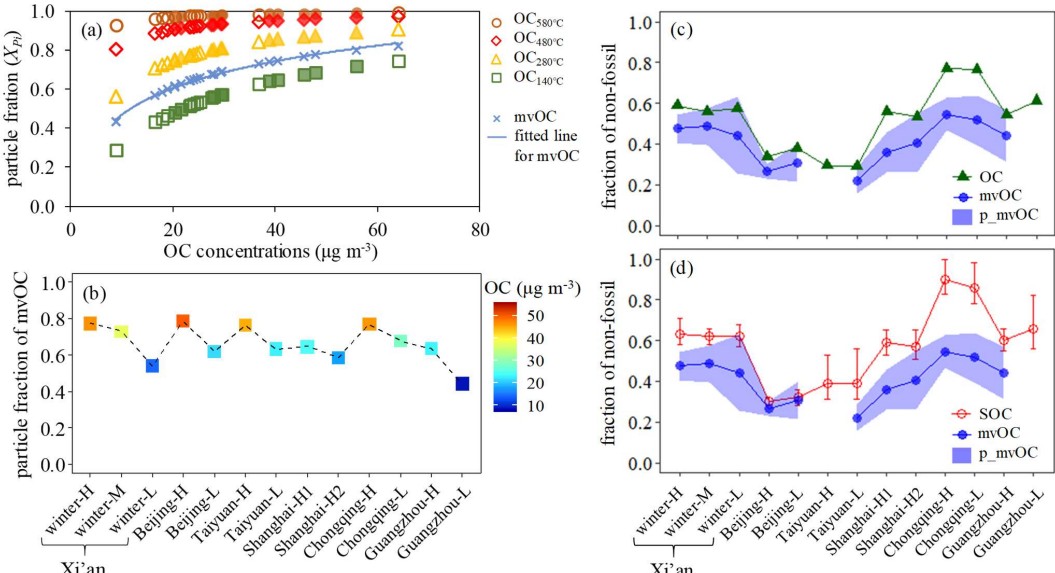

**Figure 8. (a)** The predicted particulate fraction of OC fractions (i.e., $OC_{140°C}$, $OC_{280°C}$, $OC_{480°C}$, $OC_{580°C}$, where the subscript refers to the desorption temperature of the IMPROVE_A protocol) as function of OC concentrations (Supplement S3). Hollow shapes are used to indicated samples collected by high-volume samplers (HiVol) at a flow rate of 1.0 $m^3$ $min^{-1}$ and filled shapes by mini-volume samplers (MiniVol) at a flow rate of 5 L $min^{-1}$. Particulate fraction of mvOC ($X_{p\_mvOC}$, blue cross) is estimated by interpolating the particulate fraction of $OC_{140°C}$ and $OC_{280°C}$. The blue solid line is the fitted line for the $X_{p\_mvOC}$. **(b)** The $X_{p\_mvOC}$ for each sample corresponding to the fitted line in panel a for each sample; **(c, d)** fraction of non-fossil in mvOC collected on the single bare quartz (BQ) filter ($f_{nf}$(mvOC), blue line and solid circle). The dashed area in blue indicates the possible range of the non-fossil fraction of particulate carbon after correction for the positive artifact ($f_{nf}$(p_mvOC)), using an upper limit estimate for the positive artifact. The upper and lower limits of the range correspond to the extreme assumption that the $F^{14}C$ of the adsorbed organic vapor is 50 % lower and 50 % higher than measured $F^{14}C_{(mvOC)}$, respectively. $f_{nf}$(p_mvOC) are compared with $f_{nf}$(OC) (green solid triangle) in panel c and $f_{nf}$(SOC) (red circle) in panel d. Interquartile range (25th-75th percentile) of the median $f_{fossil}$ (SOC) are shown as vertical bars.