# Peer review of "High contributions of fossil sources to more volatile organic aerosol"

_Atmospheric Chemistry and Physics, 2018_

## Referee Comment (RC1) · Anonymous Referee #2 · 18 Apr 2019

In this manuscript, the authors report their results of an extensive measurement and analysis effort to assess the contribution of fossil sources to a fraction of particulate organic carbon, i.e., more volatile organic carbon, which they define as the fraction of organic carbon desorbing at 200°C in a helium stream. The combination of numerous analytical techniques, including $^{14}$C analysis and $\delta^{13}$C determination with statistical analysis is intriguing. The authors also discuss several aspects of data uncertainty, including e.g. positive sampling artifacts due to the adsorption of volatile organic compounds onto the quartz filters used for sampling and recoveries of mvOC during filter aliquot desorption. They also outline the influence of nuclear bomb tests in the 1960s and 1970s on the measured $^{14}$C/$^{12}$C ratios. Further positive points are the detailed presentation of assumptions and estimations going into the calculations, and the ex-

tensive supporting information. The scientific methods and assumptions are presented clearly and appear to be valid. In my opinion, the presented, well-written manuscript meets all requirements for publication in ACP. However, I am not that familiar with the term 'more volatile' organic carbon and thus I do not know how common it really is and if the title is self-explanatory in its current state.

Technical comments:
1) Please check the manuscript again for the proper introduction of abbreviations before they are used for the first time
2) Page 6, Line 12: You have diluted the $CO_2$ with He, thus it should read "$CO_2/He$" mixture, not "$CO_2/H_2$" mixture

---

## Referee Comment (RC2) · Soenke Szidat (Referee) · 6 Jun 2019

The manuscript describes a novel approach of radiocarbon source apportionment, which investigates the contributions of fossil vs. non-fossil emissions to the more volatile organic carbon (mvOC) fraction that evaporates in helium at 200°C. This approach shows for particulate matter samples from different cities in China that the fossil impact on mvOC is larger than for total OC and secondary OC. This new insight has implications on a better understanding of sources of carbonaceous aerosols, which is currently a hot topic in atmospheric chemistry and physics. Therefore, I recommend accepting the manuscript after minor revisions.

Main comments:

1. P4, L1-2, P12, L18-19, P18, L10-11: In previous work (Agrios et al., 2017; https://doi.org/10.1017/RDC.2016.88), we established a continuous-flow coupling of the Sunset OC/EC analyzer with the MICADAS and measured $^{14}$C online for low-temperature OC steps and even monitored the change of the $^{14}$C signal during the temperature ramp. We also found a more fossil signal for the 200°C step than for the following OC steps using higher temperature. We furthermore observed that even a shift from fossil to non-fossil emissions occurred within the 200°C peak for some samples, which indicates that the fossil character of mvOC would probably have been even more pronounced, if Ni et al. had chosen a temperature lower than 200°C.

2. P10, L22-24: Fig. 1b reveals that low recoveries are associated with a bias in $F^{14}C$(mvOC). Therefore, the correction of sample winter-L should address this bias accordingly by subtraction. A simple increase of the uncertainty is not appropriate.

3. P11, L1-2 and P12, L8-11: In order to identify reasons for differences between cities and individual samples, meteorological data should be shown in the Supplement.

4. P13, L16-18: The fact that the correlation between $f_{nf}$(mvOC) and $f_{nf}$(OC) is better than the correlation between $f_{nf}$(mvOC) and $f_{bb}$(EC) is mainly caused by the comparison of different fractions of the carbonaceous aerosol: in the former case, two OC fractions are compared (i.e. mvOC and OC), whereas in the latter case, OC and EC are compared. $EC_{bb}$ may be transferred into $POC_{bb}$ (see Eq. 8), but one has to take into account that the large variability of rbb contributes to the r2 value of the correlation between $f_{nf}$(mvOC) and $f_{bb}$(EC) in Fig. 5. (The uncertainty of $r_{bb}$ was estimated to be 25%; see P8, L12.) Therefore, the suggestion of the importance of secondary formation of mvOC and/or other non-fossil contribution to mvOC besides primary biomass burning is not valid. This sentence should be removed. Consequently, the corresponding sentence P18, L30 to P19, L2 should also be deleted.

5. P15, L1-14: The authors try to draw conclusions from different $r^2$ values of correlations between $mvOC_{nf}$ with $POC_{bb}$ and with $SOC_{nf}$ (Fig. 7b). However, the statistical difference of both $r^2$ values was not proven by a proper test (e.g. an F-test). Furthermore, the high uncertainties of $POC_{bb}$ (see my comment to P13, L16-18) and $SOC_{nf}$ (which are indicated in Fig. 6) are also not considered for this discussion. As these important factors were not taken into account, the whole passage (P15, L1-14) should be removed.

Technical comments:

6. P1, L20: Better use the following phrasing: (range: 7 %–25 %)

7. P2, L4-7: The focus of this sentence should be changed, because a) PAHs are only a minor fraction of OC and b) EC is carcinogenic as well. I suggest characterizing OC and EC very broadly without mentioning health effects or special substance classes.

8. P2, L32: A comma is missing before "can"

9. P3, L1-3: Examples of high-volatility BBOA components should be given, as these may be relevant for the mvOC fraction.

10. P4, L10-11: pre-baked

11. P5, L29-30: Even though the blank is small compared to the sample amount, a blank correction has to be performed for both mvOC concentrations and their $F^{14}C$ values. If a direct analysis of the $F^{14}C$ of the blank hasn't been performed, a value of $0.50\pm0.29$ should be applied to cover the full $F^{14}C$ range from 0 to 1 based on the assumption that a continuous uniform distribution (i.e. a rectangular distribution) is valid.
12. P6, L7-8: Here, the same applies as for P5, L29-30.
13. P6, L22: "and 1970s" should be deleted.
14. P7-9, Chapter 2.4: In the explanation of the calculation "can be" should be substituted by "was" several times.
15. P7, L26-27: A reference should be shown for the statement that the contributions from plant detritus, bioaerosols and spores to $PM_{2.5}$ are likely small.
16. P10, L14: "the" was erroneously repeated at the beginning of the line.
17. P10, L20: "Taken together" should be removed.
18. P13, L7: I suggest to begin the sentence with "As $f_{nf}$(mvOC) is smaller"
19. P13, L14: The citation "(Fig. 4b)" should be moved to the end of the sentence in L10.
20. P15, L19: we conclude (remove "can")
21. P15, L20-21: In Fig. 7c there are two outlier data points from sample
22. P15, L30: In other words
23. P18, L3: Consequently, our conclusion
24. P18, L30: References to the literature (Masalaite et al., 2017, 2018) and figures (Fig. 5) from the paper should be removed from the Conclusions.
25. Fig. 2: An uncertainty of the average $F^{14}C$(mvOC) should be given in line 6 using the standard deviation of the three replicates.
26. Fig. 3: The following sentence should be added to the caption: "For details see Tab. S4."
27. Supplement PS4, second to last line: $OC_{280°C}$ (instead of $OC_{2800°C}$)
28. Fig. S5, last line of the caption: The panels (a) and (b) have
29. Table S4: Uncertainties are missing for $d^{13}C_{EC}$ (last column)

---

## Author Comment (AC1) · 16 Jul 2019

July 16, 2019

Dear Dr. Rupert Holzinger,

Thank you for providing us the opportunity to revise and improve our manuscript entitled "High contributions of fossil sources to more volatile organic carbon" by Ni et al. We are also grateful for the valuable comments and suggestions by the two reviewers.

The comments on the main text and supplement are addressed accordingly. Detailed responses to each of the reviewer's comments are provided in blue. Attached please also find the marked-up manuscript to track the changes in the revised manuscript.

Sincerely,

Dr. Haiyan Ni

Centre for Isotope Research (CIO)

Energy and Sustainability Research Institute Groningen (ESRIG)

University of Groningen

Nijenborgh 6, 9747 AG Groningen, the Netherlands

Email: h.ni@rug.nl

**Response to reviewer #2:**

**General comments**

In this manuscript, the authors report their results of an extensive measurement and analysis effort to assess the contribution of fossil sources to a fraction of particulate organic carbon, i.e., more volatile organic carbon (mvOC), which they define as the fraction of organic carbon desorbing at 200 °C in a helium stream. The combination of numerous analytical techniques, including $^{14}$C analysis and $^{13}$C determination with statistical analysis is intriguing. The authors also discuss several aspects of data uncertainty, including e.g. positive sampling artifacts due to the adsorption of volatile organic compounds onto the quartz filters used for sampling and recoveries of mvOC during filter aliquot desorption. They also outline the influence of nuclear bomb tests in the 1960s and 1970s on the measured $^{14}$C/$^{12}$C ratios. Further positive points are the detailed presentation of assumptions and estimations going into the calculations, and the extensive supporting information. The scientific methods and assumptions are presented clearly and appear to be valid. In my opinion, the presented, well-written manuscript meets all requirements for publication in ACP.

**Response:** We thank the reviewer for the nice summary of our manuscript and the positive evaluation of our work. We have carefully addressed the reviewer's comments. Below are point-to-point responses.

**Specific comments**

However, I am not that familiar with the term 'more volatile' organic carbon and thus I do not know how common it really is and if the title is self-explanatory in its current state.

**Response:** In this manuscript, mvOC is operationally defined as more volatile fraction of total organic carbon (OC) aerosol that evaporates in helium at 200 °C. The term "more volatile" OC refers to the more volatile OC fraction compared to the total OC. This is clarified several times in the Introduction section, for example:

> "In some recent studies, a thermal desorption approach has been used for analysis of filter samples at lower and higher temperatures as an indicator of volatility (Holzinger et al., 2013; Lopez-Hilfiker et al., 2014, 2015; Timkovsky et al., 2015; Masalaite et al., 2017, 2018). Even though the desorption temperature is not a direct measure of the particle volatility, OC desorbed from the filter at lower temperature tends to be more volatile, whereas the less volatile (i.e., more refractory) OC tends to be desorbed at higher temperatures." (page 3, line 16-20)

> "In addition, we operationally define a more volatile OC fraction (mvOC) as the carbon fraction desorbed from the filter samples at 200 °C in helium (He), and investigate sources of mvOC." (page 4, line 11-12)

In the literature, the term "more volatile" (or "less refractory") and "less volatile" (or "more refractory") are usually used, when thermal desorption is used for separation of different fractions of organic aerosol (e.g., Masalaite et al., 2017, 2018; Meusinger et al., 2017), or thermodenuder

(TD) is used for determination of volatility of organic aerosols (e.g., Huffman et al., 2008, 2009; Zhang et al., 2011).

Following the reviewer's concern about the title, we change the title to better reflect the content of the manuscript and to be more specific (changes are underlined):

"High contributions of fossil sources to more volatile organic aerosol"

**Technical comments**

**1)** Please check the manuscript again for the proper introduction of abbreviations before they are used for the first time.

**Response:** Thank you for this comment. We have checked the abbreviations accordingly and defined abbreviations when first used. The following abbreviations are defined in the revised manuscript:

"mvOC is isolated by desorbing organic carbon from the filter samples in helium (He) at 200 °C" (page 1, line 15)

"Secondary OC (SOC) concentrations and sources are modelled based on the $^{14}$C-apportioned OC and EC" (page 1, line 22)

"In this study, wintertime fine particulate matter (PM$_{2.5}$, particles with aerodynamic diameter < 2.5 μm) samples from 6 Chinese megacities are studied." (page 4, line 8-9)

"filter pieces of 1.5 cm$^2$ were taken for OC and EC analysis using a carbon analyzer (Model 5L, Sunset Laboratory, Inc., Portland, OR, USA) following the thermal-optical transmittance protocol EUSAAR_2 (European Supersites for Atmospheric Aerosol Research; Cavalli et al., 2010)." (page 4, line 29 to page 5, line 1)

"Different from EUSAAR_2 protocol, IMPROVE_A (Interagency Monitoring of Protected Visual Environments) protocol defines OC fractions at stepwise temperature of 140 °C, 280 °C, 480 °C, and 580 °C in He." (page 5, line 8-10)

"we selected the samples carefully to cover periods of low and high total carbon (TC, the sum of OC and EC) and PM$_{2.5}$ concentrations to get samples representative of various pollution conditions that did occur in each city." (page 5, line 14-15)

**2)** Page 6, Line 12: You have diluted the CO$_2$ with He, thus it should read "CO$_2$/He"

mixture, not "CO$_2$/H$_2$" mixture

**Response:** Thank you for your carefully reading. Done. Now it reads:

"The CO$_2$/He mixture is directly fed into the Cs sputter ion sources of…" (page 6, line 30)

**References:**

Huffman, J., Docherty, K., Mohr, C., Cubison, M., Ulbrich, I., Ziemann, P., Onasch, T., and Jimenez, J.: Chemically-resolved volatility measurements of organic aerosol from different sources, Environ. Sci. Technol., 43, 5351-5357, 2009.

Huffman, J. A., Ziemann, P. J., Jayne, J. T., Worsnop, D. R., and Jimenez, J. L.: Development and characterization of a fast-stepping/scanning thermodenuder for chemically-resolved aerosol volatility measurements, Aerosol Sci. Tech., 42, 395-407, 2008.

Masalaite, A., Holzinger, R., Remeikis, V., Roeckmann, T., and Dusek, U.: Characteristics, sources and evolution of fine aerosol ($PM_1$) at urban, coastal and forest background sites in Lithuania, Atmos. Environ., 148, 62-76, 2017.

Masalaite, A., Holzinger, R., Ceburnis, D., Remeikis, V., Ulevičius, V., Röckmann, T., and Dusek, U.: Sources and atmospheric processing of size segregated aerosol particles revealed by stable carbon isotope ratios and chemical speciation, Environm. Pollut., 240, 286-296, https://doi.org/10.1016/j.envpol.2018.04.073, 2018.

Meusinger, C., Dusek, U., King, S. M., Holzinger, R., Rosenørn, T., Sperlich, P., Julien, M., Remaud, G. S., Bilde, M., Röckmann, T., and Johnson, M. S.: Chemical and isotopic composition of secondary organic aerosol generated by $\alpha$-pinene ozonolysis, Atmos. Chem. Phys., 17, 6373-6391, https://doi.org/10.5194/acp-17-6373-2017, 2017.

Zhang, Q., Jimenez, J. L., Canagaratna, M. R., Ulbrich, I. M., Ng, N. L., Worsnop, D. R., and Sun, Y.: Understanding atmospheric organic aerosols via factor analysis of aerosol mass spectrometry: a review, Anal. Bioanal. Chem., 401, 3045-3067, 10.1007/s00216-011-5355-y, 2011.

**Response to reviewer #3:**

**General comments**

The manuscript describes a novel approach of radiocarbon source apportionment, which investigates the contributions of fossil vs. non-fossil emissions to the more volatile organic carbon (mvOC) fraction that evaporates in helium at 200°C. This approach shows for particulate matter samples from different cities in China that the fossil impact on mvOC is larger than for total OC and secondary OC. This new insight has implications on a better understanding of sources of carbonaceous aerosols, which is currently a hot topic in atmospheric chemistry and physics. Therefore, I recommend accepting the manuscript after minor revisions.

**Response:** Thank you, Sönke Szidat, for the helpful comments and providing us the opportunity to strengthen our research. We try to address all of them carefully and this is really a big help to improve the manuscript.

**Specific comments**

Main comments:

**1)** P4, L1-2, P12, L18-19, P18, L10-11: In previous work (Agrios et al., 2016; https://doi.org/10.1017/RDC.2016.88), we established a continuous-flow coupling of the Sunset OC/EC analyzer with the MICADAS and measured $^{14}$C online for low temperature OC steps and even monitored the change of the $^{14}$C signal during the temperature ramp. We also found a more fossil signal for the 200°C step than for the following OC steps using higher temperature. We furthermore observed that even a shift from fossil to non-fossil emissions occurred within the 200°C peak for some samples, which indicates that the fossil character of mvOC would probably have been even more pronounced, if Ni et al. had chosen a temperature lower than 200°C.

**Response:** We regret that we were not aware of the work by Agrios et al. (2016), and thank you for pointing this out. We read Agrios et al. (2016) very carefully and find that the low temperature OC steps including 200°C step were conducted in $O_2$. However, in this study mvOC is desorbed at 200°C in helium (He). OC extracted in $O_2$ and He even at the same temperature can be very different in the extracted carbon mass and therefore $F^{14}C$ values. Thus, OC extracted at 200°C in $O_2$ by Agrios et al. (2016) can not be compared directly to mvOC defined in this study. But Angrios et al. (2016) and this study have the qualitatively similar finding that OC extracted at 200°C is more fossil than that extracted at higher temperature, no matter the extraction occurs in $O_2$ or He.

However, we do not know if we can extrapolate this conclusion to a temperature lower than 200 °C. Agrios et al. (2016) observed a shift from fossil to non-fossil emissions occurred within the 200°C peak for some samples extracted in $O_2$, indicating OC extracted at temperature lower than 200°C is more fossil than OC at 200°C. In contrast to Agrios et al. (2016), we extracted mvOC in He and observed higher $F^{14}C$ (i.e., less fossil) for samples with mvOC recovery <1, which probably resulted from lower desorption temperature than 200°C, as discussed in the second paragraph of Sect. 3.1 and shown in Fig. 1b.

We think this citation is very helpful for readers to understand the fossil character of OC extracted at lower temperature, and we add the citation in the revised text.

"Agrios et al. (2016) had the qualitatively similar finding that $F^{14}C$ of OC extracted in $O_2$ at 200°C was smaller than $F^{14}C$ of OC extracted in $O_2$ at higher temperature for samples collected from an urban and rural site in the Switzerland and from the Los Angeles Basin, USA." (page 13, line 10-12)

Furthermore, we avoid using the expression "*the first time* that the source contribution to more volatile fraction of OC" or "*the first $^{14}C$ measurement* of the more volatile fraction of OC". The revised text shows (changes are underlined):

"In addition, we operationally define a more volatile OC fraction (mvOC) as the carbon fraction desorbed from the filter samples at 200 °C in helium (He), and investigate sources of mvOC. The source contribution to mvOC is measured unambiguously by $^{14}C$." (page 4, line 11-14)

"We can unambiguously conclude from $^{14}C$ measurements of mvOC and OC that mvOC is more fossil than OC in six Chinese cites." (page 13, line 13-14)

"Radiocarbon measurements are conducted for EC and OC of $PM_{2.5}$ samples collected in six Chinese megacities at wintertime. In addition, the sources of mvOC are measured unambiguously using $^{14}C$." (page 19, line 4-6)

**2)** P10, L22-24: Fig. 1b reveals that low recoveries are associated with a bias in $F^{14}C_{(mvOC)}$. Therefore, the correction of sample winter-L should address this bias accordingly by subtraction. A simple increase of the uncertainty is not appropriate.

**Response:** Fig. 1b shows that for sample winter-H and winter M, low recoveries of mvOC lead to biased high $F^{14}C_{(mvOC)}$ values. But we are not sure if we can generalize this conclusion to other samples. In addition, Agrios et al. (2016) found that for some samples a shift from fossil to non-fossil emissions occurred within the 200°C peak, which indicates that opposite to Fig. 1b, low recovery of mvOC could in principle also result in biased low $F^{14}C_{(mvOC)}$. The findings in Agrios et al. (2016) and Fig. 1b of this study suggest that low recovery of mvOC can lead to either biased higher or lower $F^{14}C_{(mvOC)}$ values.

In this study, the mvOC recovery of sample winter-L is low (0.51), and we could not repeat it due to due to the limited filter material. Instead, we take the measured $F^{14}C_{(mvOC)}$ for sample winter-L but assign a bigger absolute uncertainty of 0.05, due to its low mvOC recovery. This is based on the difference in $F^{14}C_{(mvOC)}$ for winter-H and winter-M with low and high mvOC recoveries which is roughly 0.05 (Fig. 1b), as explained in the Sect. 3.1. The larger uncertainty is assigned for the measured $F^{14}C_{(mvOC)}$ for sample winter-L so that possible effects of its low mvOC recovery are taken into consideration and could be either positive or negative

**3)** P11, L1-2 and P12, L8-11: In order to identify reasons for differences between cities and individual samples, meteorological data should be shown in the Supplement.

**Response:** Meteorological data including wind speed, temperature and relative humidity (RH) are added in Supplemental Table S1. There was no precipitation in all cities during all sampling dates, since we only included samples collected on dry days in this study, to eliminate this variable as a potential confounding factor.

TC concentrations in Chongqing were higher than the other southern Chinese cities (Shanghai and Guangzhou) (P11, L1-2 in the original manuscript). Beside high anthropogenic emissions, higher TC concentrations in Chongqing were associate with unfavorable meteorological conditions characterized by high RH and low wind speed, which enhanced both the accumulation of pollutants and the formation of secondary aerosol (Zheng et al., 2015; Tie et al., 2017).

For samples collected in the same city, we found that compared to samples representing low (L) TC concentrations, samples representing high (H) TC concentrations were collected under more stagnant conditions indicated by lower wind speed (Table S1 in the revised manuscript).

4) P13, L16-18: The fact that the correlation between $f_{nf}$(mvOC) and $f_{nf}$(OC) is better than the correlation between $f_{nf}$(mvOC) and $f_{bb}$(EC) is mainly caused by the comparison of different fractions of the carbonaceous aerosol: in the former case, two OC fractions are compared (i.e. mvOC and OC), whereas in the latter case, OC and EC are compared. $EC_{bb}$ may be transferred into $POC_{bb}$ (see Eq. 8), but one has to take into account that the large variability of $r_{bb}$ contributes to the $r^2$ value of the correlation between $f_{nf}$(mvOC) and $f_{bb}$(EC) in Fig. 5. (The uncertainty of $r_{bb}$ was estimated to be 25%; see P8, L12.) Therefore, the suggestion of the importance of secondary formation of mvOC and/or other non-fossil contribution to mvOC besides primary biomass burning is not valid. This sentence should be removed. Consequently, the corresponding sentence P18, L30 to P19, L2 should also be deleted.

**Response:** We agree with the reviewer that correlation between $f_{nf}$(mvOC) and $f_{nf}$(OC) is better than the correlation between $f_{nf}$(mvOC) and $f_{bb}$(EC) is mainly caused by comparison of two OC fractions in the former case. In addition, as the reviewer explained, the very variable $r_{bb}$ (i.e., $POC_{bb}/EC_{bb}$ ratio) and the unknown mvOC/EC ratio from primary biomass burning can contribute to better correlation between $f_{nf}$(mvOC) and $f_{nf}$(OC) than that between $f_{nf}$(mvOC) and $f_{bb}$(EC).

Consequently, we have deleted the following sentence (a strikethrough is used to indicated the deletion of text) as the reviewer suggested:

> "" (page 14, line 11-13)

> "" (page 19, line 26-29)."

5) P15, L1-14: The authors try to draw conclusions from different $r^2$ values of correlations between $mvOC_{nf}$ with $POC_{bb}$ and with $SOC_{nf}$ (Fig. 7b). However, the statistical difference of both $r^2$ values was not proven by a proper test (e.g. an F-test). Furthermore, the high uncertainties of $POC_{bb}$ (see my comment to P13, L16-18) and $SOC_{nf}$ (which are indicated in Fig. 6) are also not considered for this discussion. As these important factors were not taken into account, the whole passage (P15, L1-14) should be removed.

**Response:** To statistically compare the $r^2$ values of correlations between $mvOC_{nf}$ with $POC_{bb}$ and with $SOC_{nf,}$, we compute their bootstrap $r^2$ values and then evaluate overlap of the corresponding 95% confidence intervals. The bootstrapped 95% confidence intervals for $r^2$ of correlations between $mvOC_{nf}$ with $POC_{bb}$ and with $SOC_{nf}$ are (0.21, 0.94) and (0.60, 0.99), respectively. Their 95% confidence intervals are clearly overlapped, therefore there is no significant difference in $r^2$ values of correlations between $mvOC_{nf}$ with $POC_{bb}$ and with $SOC_{nf}$. We chose the bootstrap method as opposed to a parametric method, due to the few data points in the regression, which made it difficult to determine, if the residuals have a normal distribution, i.e. if a parametric test is applicable.

Furthermore, we agree with the reviewer that the high uncertainties of $POC_{bb}$ and $SOC_{nf}$ that resulted from the uncertain $r_{bb}$ should be considered for this discussion. Consequently, we rewrite the passage (page 15, line 27 to page 16, line 9):

> "Good correlation was found between $mvOC_{nf}$ and $SOC_{nf}$ ($r^2 = 0.87$, bootstrapped 95% confidence interval: (0.21, 0.94); Fig. 7), which is not significantly different from the correlation between $mvOC_{nf}$ and $POC_{bb}$ ($r^2 = 0.71$, bootstrapped 95% confidence interval: (0.60, 0.99)). $SOC_{nf}$ is estimated by subtracting primary OC from biomass burning ($POC_{bb}$) from $OC_{nf}$, where $POC_{bb}$ is estimated by multiplying $EC_{bb}$ with the OC/EC ratio of fresh biomass burning plumes ($r_{bb}$) (Eq. 7). Due to the large uncertainty in $r_{bb}$, separation between $POC_{bb}$ and $SOC_{nf}$ is rather uncertain, and the $r^2$ for a correlation of $mvOC_{nf}$ and $SOC_{nf}$ is therefore affected by $r_{bb}$. For example, with lower $r_{bb}$ of 3 (mean), the correlation between $SOC_{nf}$ and $mvOC_{nf}$ is stronger ($r^2 = 0.92$) than for our best estimate of $r_{bb}$ (4) ($r^2 = 0.87$). For $r_{bb}$ of 5, it is slightly weaker ($r^2 = 0.76$, Fig S8). $OC_{nf}$, the sum of $POC_{bb}$ and $SOC_{nf}$, is well-constrained by $F^{14}C_{OC}$ and not affected by $r_{bb}$. $mvOC_{nf}$ correlates strongly with $OC_{nf}$ ($r^2 = 0.91$), suggesting strong impacts on the variability of $mvOC_{nf}$ from non-fossil emissions including the secondary formation from non-fossil sources and primary biomass burning emissions."

Consequently, the corresponding sentences in the Conclusion section are also revised:

> "Good correlation is found between $mvOC_{nf}$ and $OC_{nf}$ ($r^2 = 0.91$). This indicates strong impacts on the variability of $mvOC_{nf}$ from non-fossil emissions, including the secondary formation of OC from non-fossil sources and primary biomass burning." (page 20, line 6-9)

**Technical comments:**

**6)** P1, L20: Better use the following phrasing: (range: 7 %–25 %)

**Response:** Thank you for pointing this out. Done. The revised abstract shows:

> "The average difference in the fossil fractions between mvOC and OC is 13 % (range: 7 %–25 %)," (page 1, line 21)

There are also several occasions in the Results/Conclusion sections, and the revised text shows:

> "mvOC and OC concentrations averaged 3.3 ± 2.2 µg m$^{-3}$ (range: 0.7–7.4 µg m$^{-3}$) and…" (page 11, line 19)

> "The relative contribution of fossil sources to mvOC is on average 59 % (range: 45 %–78 %)," (page 20, line 5)

**7)** P2, L4-7: The focus of this sentence should be changed, because a) PAHs are only a minor fraction of OC and b) EC is carcinogenic as well. I suggest characterizing OC and EC very broadly without mentioning health effects or special substance classes.

**Response:** Thank you for this comment. The revised manuscript shows:

> "Aerosol particles are of importance for atmospheric chemistry and physics, and exert a crucial effect on the climate system, air quality and human health (Fuzzi et al., 2015). The carbonaceous fraction of aerosol comprises a large fraction of the fine aerosol mass (20 %–80 %) (Cao et al., 2007; Tao et al., 2017). Carbonaceous aerosols are often operationally subdivided into organic carbon (OC) and elemental carbon (EC). OC consists of a large variety of organic species that cover a great range of volatilities and are not or weakly light absorbing, while EC is non-volatile, resistant to chemical transformation and strongly light absorbing (Pöschl, 2005)." (page 2, line 2-10)

**8)** P2, L32: A comma is missing before "can"

**Response:** The revised manuscript shows:

> "the volatility of different OA components, such as hydrocarbon-like OA (HOA), biomass burning OA (BBOA) and OOA, can be estimated for ambient aerosols." (page 3, line 2-4)

**9)** P3, L1-3: Examples of high-volatility BBOA components should be given, as these may be relevant for the mvOC fraction.

**Response:** Following the reviewer's suggestion, we do the literature search of high-volatility BBOA components. To the best of our knowledge, studies on the volatility of BBOA components are very limited. When BBOA is identified and quantified using aerosol mass spectrometry (AMS) technique in combination with positive matrix factorization (PMF), the detection and identification of specific molecular tracers is not possible. This is because the AMS technique

utilizes electron impact ionization resulting in extensive fragmentation. However, the volatility distribution for BBOA components can be estimated using the volatility basis set. For example, Paciga et al. (2016) found that the BBOA in an urban background site in Paris contained 50% semi-volatile organic components (SVOCs with effective saturation concentrations $C^*$ of 1–100 µg m$^{-3}$ range) and 30% low volatility organic compounds (LVOCs with $C^*$ of 10$^{-3}$–0.1 µg m$^{-3}$).

Recently, high-resolution-time-of-flight chemical ionization mass spectrometer (HRTOF-CIMS) coupled to a Filter Inlet for Gases and AEROsols (FIGAERO) is developed (Lopez-Hilfiker et al., 2014) and used for measurements of ambient biomass burning aerosol (e.g., Gaston et al., 2016). This technique allows the detection and quantification of oxygenated and nitrogen-containing compounds on the molecular level and provides information on volatility by detecting gas-and particle-phase compounds simultaneously. Gaston et al. (2016) applied this technique to BBOA in the northwestern US. They found that levoglucosan (i.e., the biomass burning tracer) is semi-volatile and has the potential to form low volatility products during the chemical aging of ambient biomass burning aerosol.

Regarding the high-volatility BBOA components, the following underlined sentences have been added in the Introduction section:

> "Source and ambient studies indicate that BBOA and HOA are generally more volatile than OOA. Meanwhile, the volatility of BBOA can be quite variable, depending on type of biomass and the combustion conditions, and either higher or lower than that of HOA (e.g., Grieshop et al. 2009c; Huffman et al. 2009a; Paciga et al. 2016; Cao et al., 2018). For example, Huffman et al. (2009b) measured the volatility of primary OA from biomass burning and found that the more volatile BBOA was generally dominated by smoldering combustion, while the less volatile BBOA was more influenced by flaming combustion. Paciga et al. (2016) found that the BBOA was more volatile than HOA in a urban background site in Paris, with 50% semi-volatile organic components (SVOCs with effective saturation concentrations $C^*$ of 1–100 µg m$^{-3}$ range) and 30% low volatility organic compounds (LVOCs with $C^*$ of 10$^{-3}$–0.1 µg m$^{-3}$). Gaston et al. (2016) found that levoglucosan (i.e., the biomass burning tracer) was semi-volatile and had the potential to form low volatility products during the chemical aging of ambient biomass burning aerosol, applying high-resolution-time-of-flight chemical ionization mass spectrometer (HRTOF-CIMS) coupled to a Filter Inlet for Gases and AEROsols (FIGAERO) to aerosol in the northwestern US. " (page 3, line 4-15)

**10)** P4, L10-11: pre-baked

**Response:** Corrected. The revised manuscript shows:

> "PM$_{2.5}$ samples were collected on pre-baked (780 °C for 3 h) quartz fiber filter…" (page 4, line 22)

**11)** P5, L29-30: Even though the blank is small compared to the sample amount, a blank correction has to be performed for both mvOC concentrations and their F$^{14}$C values. If a direct

analysis of the $F^{14}C$ of the blank hasn't been performed, a value of $0.50 \pm 0.29$ should be applied to cover the full $F^{14}C$ range from 0 to 1 based on the assumption that a continuous uniform distribution (i.e. a rectangular distribution) is valid.

**Response:** Thank you for pointing this out.

(1) blank corrections for mvOC concentrations

The contamination introduced by the combustion processes yields $0.52 \pm 0.31$ µgC mvOC, which was determined by following exactly the same mvOC isolating procedures with either empty filter boat or with blank filters. Blank corrections for mvOC mass from the isolation procedure were performed in the original manuscript, but we did not state this clearly. In the revised manuscript, we add:

> "The contamination introduced by the combustion process yields $0.72 \pm 0.44$ µgC EC, $0.85 \pm 0.49$ µgC OC, $0.52 \pm 0.31$ µgC mvOC per extraction, respectively. Compared with our sample size of 30–391 µgC EC, 30–445 µgC OC, and 15–121 µgC mvOC, the blanks are $\leq 7$ % of the mvOC sample amount and < 3% of OC and EC sample amount, therefore relatively small compared to our sample sizes. The mvOC concentrations reported in this study are corrected for contamination by subtraction." (page 6, line 7-11)

(2) blank corrections for $F^{14}C_{(mvOC)}$

The $^{14}C$ values of carbon fractions (i.e., mvOC, OC and EC) can be blank corrected according to the mass balance equation:

$$F^{14}C_S = \frac{F^{14}C_m \times M_m - F^{14}C_b \times M_b}{M_m - M_b} \qquad (S4)$$

where $F^{14}C_S$ is $F^{14}C$ of the aerosol carbon collected on the filter, which is blank corrected, $F^{14}C_m$ and $M_m$ are the measured $F^{14}C$ and the measured mass of the respective carbon fraction (mvOC, OC or EC), and $F^{14}C_b$ and $M_b$ the fraction modern and the mass of the respective carbon fraction of blanks.

Since the directly determined system blank (i.e. the amount of $CO_2$ measured, when no sample is introduced into the aerosol combustion system) is $0.52 \pm 0.31$ µgC mvOC per extraction (i.e., $M_b$ for mvOC), it is much too small to be analyzed for $F^{14}C_b$ of mvOC. Therefore, the $F^{14}C_{(mvOC)}$ values in the manuscript were not blank corrected.

To do blank corrections for mvOC, we need to estimate its $F^{14}C_b.$. We assume that $F^{14}C_b$ for mvOC ranges from 0 to 1 with a continuous uniform distribution as the reviewer suggested. To propagate uncertainties, a Monte Carlo simulation with 10,000 individual calculations of $F^{14}C_S$ for mvOC was conducted according to Eq. (S4). For calculation inputs, $F^{14}C_b$ of mvOC. was randomly chosen from a continuous uniform distribution between 0 and 1. $F^{14}C_m$ for mvOC was randomly chosen from a normal distribution symmetric around the measured values with uncertainties as standard deviation (Table S4). The random values for $M_m$ and $M_b$ were taken from a triangular distribution, which has its maximum at the central value and 0 at the upper and

lower limits. Then 10,000 different estimations of $F^{14}C_S$ for mvOC were calculated. The derived average represents the best estimate, and the standard deviation represents the combined uncertainties.

Figure S9 shows the $F^{14}C$ of mvOC before and after blank corrections ($F^{14}C_m$ and $F^{14}C_s$, repectively) for the contamination introduced by the isolation procedure.

[Figure]

**Figure S9.** Fraction modern ($F^{14}C$) of mvOC before and after blank corrections ($F^{14}C_m$ and $F^{14}C_s$, respectively) for the contamination introduced by the isolation procedure. Detailed method to do the blank corrections is described in Supplement S3.

As shown in Fig. S9, the differences in $F^{14}C$ values of mvOC before and after blank corrections are really small, with the biggest absolute difference of 0.009 for sample Taiyuan-L. The small correction of $F^{14}C$ for mvOC will not affect any conclusion from this study. If possible, we think it probably acceptable to show the $F^{14}C$ of mvOC before corrections as it was in the original manuscript. Otherwise, we need to recalculated everything, and change nearly all figures and tables. This will not change any conclusion but will lead to inconsistence between the discussion paper and the final one.

For OC and EC, the contamination introduced by the isolation procedure yields 0.72 ± 0.44 µgC EC and 0.85 ± 0.49 µgC OC per extraction, respectively, which are the directly determined system blank (i.e. the amount of $CO_2$ measured, when no sample is introduced into the aerosol combustion system). The blanks are less than 3% of the sample amount and thus can be neglected, compared with our sample size of 30–391 µgC EC per extraction and 30–445 µgC OC per extraction. Therefore, the blank correction will not introduce large uncertainties to the data. In this study, the contamination is assessed but not used for further data correction for $F^{14}C$ of OC and EC.

To make this clear, we add the detailed calculation of blank correction for $F^{14}C$ of mvOC, OC and EC in supplemental material (Supplement S3). Figure S9 is also added in the revised Supplement. Furthermore, the revised manuscript adds:

"Compared with our sample size of 30–391 µgC EC, 30–445 µgC OC, and 15–121 µgC mvOC, the blanks are ≤7 % of the mvOC sample amount and < 3% of OC and EC sample amount, therefore relatively small compared to our sample sizes. The mvOC concentrations reported in this study are corrected for contamination by subtraction. OC and EC concentrations are measured following thermal-optical protocols using carbon analyzers (Sect. 2.2), thus are not affect by the isolation procedure using the ACS. For $^{14}C$ values (Eq. 1), the contamination is assessed but not used for further $^{14}C$ data correction for mvOC, OC and EC, because the corrections for the small blanks will not introduce large uncertainties to the data, as explained in Supplement S3." (page 6, line 9-15)

**12)** P6, L7-8: Here, the same applies as for P5, L29-30.

**Response:** OC and EC concentrations were determined following thermal-optical protocols using carbon analyzers (Sect. 2.2, page 4), which are corrected by the system blank and thus were not affected by the isolation procedure of OC and EC using aerosol combustion system (ACS). To make this clear, we add:

"OC and EC concentrations are measured following thermal-optical protocols using carbon analyzers (Sect. 2.2), thus are not affect by the isolation procedure using the ACS." (page 6, line 11-13).

(2) blank corrections for $F^{14}C$ of OC and EC

The calculation of of blank correction for $F^{14}C$ of OC and EC is given in response of question 11 and in the Supplement S3. In question 11, the contamination is the directly determined system blank (i.e. the amount of $CO_2$ measured, when no sample is introduced into the aerosol combustion system). We have explained in the response of question 11 why the contamination is assessed but not further used for $F^{14}C$ data correction. Here, the contamination is determined indirectly by combusting small amounts of standard material, with known $F^{14}C$ values, i.e., the OXII standard ($F^{14}C$ =1.3406) and $^{14}C$ free standard ($F^{14}C$ = 0). If contamination is introduced into the combustion, the measured $F^{14}C$ values of the standards will deviate from the nominal values and from this deviation the contamination can be estimated. The contamination inferred in this indirect way is below 1.5 µgC per extraction, and this is an estimate for TC blank (TC = OC+ EC). It is also relatively small compared to the size of OC (30–445 µgC) and EC samples (30–391 µgC) in this study and thus will not lead to large uncertainties to the $F^{14}C$ data. The $F^{14}C$ of the contamination can also be indirectly inferred from the standard measurements and varied between 0.2 and 0.6 for standards measured on the system with the last two years. This is broadly within the range of our samples and implies that a correction for $F^{14}C$ will even be considerably smaller than 3%.

The revised text shows:

"The contamination inferred in this indirect way is below 1.5 μgC per extraction, which is slightly higher than the directly measured contamination of OC and EC separately but in the range of a TC contamination. It is also relatively small compared to the size of OC (30–445 μgC) and EC samples (30–391 μgC) in this study and thus can be neglected (Supplement S3). The $F^{14}C$ (Eq. 1) of the contamination can also be indirectly inferred and varied between 0.2 and 0.6 for standards measured on the ACS with the last two years. This is broadly within the range of our samples and implies that a correction for $F^{14}C$ will even be considerably smaller than 3%." (page 6, line 21-26)

**13)** P6, L22: "and 1970s" should be deleted.

**Response:** Done. The revised text shows that:

"the $F^{14}C$ values of contemporary carbon are higher than 1 due to the nuclear bomb tests that nearly doubled the $^{14}CO_2$ in the atmosphere in the 1960s" (page 7, line 8-9)

**14)** P7-9, Chapter 2.4: In the explanation of the calculation "can be" should be substituted by "was" several times.

**Response:** We correct them all in Chapter 2.4. (page 7-9)

**15)** P7, L26-27: A reference should be shown for the statement that the contributions from plant detritus, bioaerosols and spores to $PM_{2.5}$ are likely small.

**Response:** We add Song et al. (2006), Hu et al. (2010) and Guo et al. (2012). (page 8, line 14)

The new citations are included in the revised reference list:

Guo, S., Hu, M., Guo, Q., Zhang, X., Zheng, M., Zheng, J., Chang, C. C., Schauer, J. J., and Zhang, R.: Primary sources and secondary formation of organic aerosols in Beijing, China, Environ. Sci. Technol., 46, 9846–9853, 2012.

Hu, D., Bian, Q., Lau, A. K. H., and Yu, J. Z.: Source apportioning of primary and secondary organic carbon in summer $PM_{2.5}$ in Hong Kong using positive matrix factorization of secondary and primary organic tracer data, J. Geophys. Res.-Atmos., 115, 10.1029/2009jd012498, 2010.

Song, Y., Zhang, Y., Xie, S., Zeng, L., Zheng, M., Salmon, L. G., Shao, M., and Slanina, S.: Source apportionment of $PM_{2.5}$ in Beijing by positive matrix factorization, Atmos. Environ., 40, 1526-1537, https://doi.org/10.1016/j.atmosenv.2005.10.039, 2006.

**16)** P10, L14: "the" was erroneously repeated at the beginning of the line.

**Response:** Corrected. (page 11, line 6)

**17)** P10, L20: "Taken together" should be removed.

**Response:** Done. (page 11, line 12)

**18)** P13, L7: I suggest to begin the sentence with "As fnf(mvOC) is smaller"

**Response:** Done. (page 14, line 2)

**19)** P13, L14: The citation "(Fig. 4b)" should be moved to the end of the sentence in L10.

**Response:** Done. (page 14, line 5)

**20)** P15, L19: we conclude (remove "can")

**Response:** Done. (page 16, line 14)

**21)** P15, L20-21: In Fig. 7c there are two outlier data points from sample

**Response:** Done. The revised text shows

"In Fig. 7c there are two outlier data points from sample…." (page 16, line 16)

**22)** P15, L30: In other words

**Response:** We have replaced "In another word" by "In other words" (page 16, line 25)

**23)** P18, L3: Consequently, our conclusion

**Response:** We have replaced "That is" with "Consequently". The revised manuscript shows:

"Consequently, our conclusion that mvOC is more fossil than OC and SOC is still valid…" (page 18, line 28)

**24)** P18, L30: References to the literature (Masalaite et al., 2017, 2018) and figures (Fig. 5) from the paper should be removed from the Conclusions.

**Response:** Done. (page 19, line 26)

**25)** Fig. 2: An uncertainty of the average $F^{14}C_{(mvOC)}$ should be given in line 6 using the standard deviation of the three replicates.

**Response:** Done, the revised caption of Fig. 2 shows:

"The dashed line in dark green indicates the expected mvOC loading and the horizontal in solid green the $F^{14}C_{(mvOC)}$ for the combined winter-M (weighted mean $\pm$ standard deviation; $0.524 \pm 0.028$)." (page 29, line 5-6)

**26)** Fig. 3: The following sentence should be added to the caption: "For details see Tab. S4."

**Response:** Done. (page 30, line 5)

**27)** Supplement PS4, second to last line: $OC_{280°C}$ (instead of $OC_{2800°C}$)

**Response:** Corrected. The revised Supplement shows:

"The desorption temperature for mvOC is 200 °C, falling within the 140 °C for $OC_{140°C}$ and 280 °C for $OC_{280°C}$. We thus estimated the particulate fraction for mvOC will fall within the $X_{p1}$ for $OC_{140°C}$ and $X_{p2}$ for $OC_{280°C}$." (page S6 in the Supplement)

**28)** Fig. S5, last line of the caption: The panels (a) and (b) have

**Response:** Corrected. Now it reads "The panel (a) and (b) have the same *x*-axis." (page S11 in the Supplement)

**29)** Table S4: Uncertainties are missing for $\delta^{13}C_{EC}$ (last column)

**Response:** The uncertainties of $\delta^{13}C_{EC}$ have been added in Table S4. (page S19 in the Supplement)

[revised manuscript text omitted]
_{\text{nf}}(\text{mvOC})$ is smaller than $f_{\text{nf}}(\text{OC})$, this is equivalent to $(\text{mvOC/OC})_{\text{fossil}} > (\text{mvOC/OC})_{\text{nf}}$ (Eq. 13). That is, OC from fossil sources contains a larger more volatile fraction than OC from non-fossil sources. This is true for all studied cities, despite the variation of $f_{\text{nf}}(\text{mvOC})$ and $f_{\text{nf}}(\text{OC})$ in different cities. $(\text{mvOC/OC})_{\text{fossil}}$ (14 ± 6.6 %; 3.7 %–28 %) is consistently higher and

5 more variable than $(\text{mvOC/OC})_{\text{nf}}$ (7.5 ± 2.9 %; 2.6 %–11.6 %; Fig. 4b). $(\text{mvOC/OC})_{\text{fossil}}$ and $(\text{mvOC/OC})_{\text{nf}}$ in general tracks the variation of mvOC/OC, except that $(\text{mvOC/OC})_{\text{fossil}}$ for Chongqing (28% for Chongqing-H and 21% for Chongqing-L) is much higher than the other sites (averaged 12 %). When comparing the mvOC/OC within each city, it is found that $(\text{mvOC/OC})_{\text{fossil}}$ changes more strongly between H and L samples than $(\text{mvOC/OC})_{\text{nf}}$, indicated by the bigger absolute differences of $(\text{mvOC/OC})_{\text{fossil}}$ between H and L samples than that of $(\text{mvOC/OC})_{\text{nf}}$ . Correlations between

10 $f_{\text{nf}}(\text{mvOC})$ and $f_{\text{nf}}(\text{OC})$, and between $f_{\text{nf}}(\text{mvOC})$ and $f_{\text{bb}}(\text{EC})$ are examined. $f_{\text{nf}}(\text{mvOC})$ correlates more closely with $f_{\text{nf}}(\text{OC})$ ($r^2$ = 0.89) than with $f_{\text{bb}}(\text{EC})$ ($r^2$ = 0.62) (Fig. 5).

It is likely that some of mvOC are of secondary origin, since SOC is formed in the atmosphere and subsequently condenses

15 onto the aerosol particles (Hallquist et al., 2009; Jimenez et al., 2009). It has been shown that SOC formed in chambers initially desorbed at relatively low temperatures (e.g., 200 °C) (Holzinger et al., 2010; Salo et al., 2011; Meusinger et al., 2017; Gkatzelis et al., 2018). However, there is also evidence that the formation of highly oxidized low-volatile compounds can occur in SOA formation (Ehn et al., 2014). We thus compared concentrations and sources of SOC to those of mvOC. SOC concentrations and $f_{\text{fossil}}(\text{SOC})$ are estimated based on the $^{14}$C-apportioned OC and EC in combination with $p$ values of

20 0–1 (Sect. 2.4). Where $\delta^{13}$C measurements were available for high-volume filter samples (Beijing, Shanghai and Guangzhou), we compare the SOC concentrations and $f_{\text{fossil}}(\text{SOC})$ derived from $p$ values of 0–1 with $p$ values constrained by $\delta^{13}$C. In these three cities, SOC concentrations and $f_{\text{fossil}}(\text{SOC})$ without constraints of $p$ value (0–1) is very similar to that with constraints of $p$ value using $\delta^{13}$C, only with relatively larger uncertainties (Fig. 6). This indicates that the unconstrained $p$-values does not lead to significant bias in SOC concentrations and $f_{\text{fossil}}(\text{SOC})$ and the main gain in using $\delta^{13}$C is currently a decrease in

25 uncertainty.

When choosing $p$ randomly from 0 to 1, the calculated median $f_{\text{fossil}}(\text{SOC})$ is smaller than 0 for samples in Chongqing, reflecting negative $\text{SOC}_{\text{fossil}}$ values. Since $\text{SOC}_{\text{fossil}}$ is calculated by subtracting $\text{POC}_{\text{fossil}}$ from $\text{OC}_{\text{fossil}}$, this indicates $\text{POC}_{\text{
[revised manuscript text omitted]

**S1. Stable carbon isotopic composition of EC**

The stable carbon isotopic composition of EC was determined using a Finnigan MAT 251 mass spectrometer with a dual inlet system (Bremen, Germany) at the Stable Isotope Laboratory at the Institute of Earth Environment, Chinese Academy of Sciences. OC is removed by heating the filter pieces at 375 °C for 3 h in a vacuum-sealed quartz tube in the presence of CuO catalyst grains. Extraction of EC was done by heating the carbon that remained on the filters at 850 °C for 5 h. The evolved $CO_2$ from EC was isolated by a series of cold traps and quantified manometrically. The stable carbon isotopic composition of the purified $CO_2$ of EC was measured and determined as $\delta^{13}C_{EC}$. A routine laboratory working standard with a known $\delta^{13}C$ value was measured every day. The quantitative levels of $^{13}C$ and $^{12}C$ isotopes were characterized using a ratio of peak intensities of m/z 45 ($^{13}C^{16}O_2$) and 44 ($^{12}C^{16}O_2$) in the mass spectrum of $CO_2$. Samples were analyzed at least in duplicate, and all replicates showed differences less than 0.3 ‰. $\delta^{13}C$ values are reported in the delta notation with respect to the international standard Vienna Pee Dee Belemnite (V-PDB):

$$\delta^{13}C\ (‰) = \left[\frac{\left(^{13}C/^{12}C\right)_{sample}}{\left(^{13}C/^{12}C\right)_{V-PDB}} - 1\right] \times 1000. \tag{S1}$$

Details of stable carbon isotope measurements are described by our previous studies (Cao et al., 2011; Ni et al., 2018).

**S2. Separation of fossil sources into coal and vehicle for EC using Bayesian statistics**

Contributions from biomass burning and fossil sources to EC is separated by $F^{14}C_{EC}$ (Sect. 2.4). The relative contribution of fossil fuel combustion to EC ($f_{fossil}(EC)$) and biomass burning to EC ($f_{bb}(EC)$) constrained by $F^{14}C_{EC}$ is shown in Table S5. $f_{fossil}(EC)$ can be further separated into the fraction of EC contributed by coal combustion($f_{coal}$) and vehicle emissions ($f_{vehicle}$) using $\delta^{13}C$ of EC in a Bayesian Markov chain Monte Carlo (MCMC) scheme:

$$\delta^{13}C_{EC} = \delta^{13}C_{bb} \times f_{bb}(EC) + \delta^{13}C_{coal} \times f_{coal} + \delta^{13}C_{vehicle} \times f_{vehicle}, \tag{S2}$$

$$f_{coal} + f_{vehicle} = f_{fossil}(EC), \tag{S3}$$

where $\delta^{13}C_{bb}$, $\delta^{13}C_{vehicle}$ and $\delta^{13}C_{coal}$ are the $\delta^{13}C$ signature of EC emitted from biomass burning, vehicle emissions and coal combustion, respectively. The means and the standard deviations for $\delta^{13}C_{bb}$ (-26.7 ± 1.8 ‰ for C3 plants), $\delta^{13}C_{vehicle}$ (-25.5 ± 1.3 ‰) and $\delta^{13}C_{coal}$ (-23.4 ± 1.3 ‰) are established in Andersson et al. (2015) by a thorough literature search and full compilation of $\delta^{13}C$ source signatures for EC. In brief, the mean and standard deviation for $\delta^{13}C$ endmembers for the different sources are estimated as the average and standard deviation of the different data sets,

respectively.

The source endmembers for $\delta^{13}$C are less well-constrained than $f_{bb}$(EC) and $f_{fossil}$(EC), as $\delta^{13}$C varies with fuel types and combustion conditions. For example, the range of possible $\delta^{13}$C values for vehicle emission overlaps to a small extent with the range for coal combustion, although EC from vehicle emissions are usually more depleted than from coal combustion. Uncertainties of source signatures of $\delta^{13}$C (i.e., $\delta^{13}$C$_{bb}$, $\delta^{13}$C$_{vehicle}$, $\delta^{13}$C$_{coal}$), the calculated $f_{fossil}$(EC) derived from F$^{14}$C$_{EC}$, and the measured $\delta^{13}$C$_{EC}$ are propagated by the MCMC technique, where source signatures of $\delta^{13}$C introduces a larger uncertainty than $f_{fossil}$(EC) constrained by F$^{14}$C.

MCMC was implemented in the freely available R software (https://cran.r-project.org/), using the *simmr* package (https://CRAN.R-project.org/package=simmr). Convergence diagnostics were created to make sure the model has converged properly. The simulation for each sample was run with 10,000 iterations, using a burn-in of 1000 steps, and a data thinning of 100. The results of the MCMC are the posterior probability density functions (PDF) for the relative contributions from the sources (Fig. S3).

**S3. Blank corrections for F$^{14}$C of carbon fractions**

The $^{14}$C values of carbon fractions (i.e., mvOC, OC and EC) can be blank corrected accordioning to the mass balance equation:

$$F^{14}C_S = \frac{F^{14}C_m \times M_m - F^{14}C_b \times M_b}{M_m - M_b}$$
(S4)

where F$^{14}$C$_S$ is F$^{14}$C of the aerosol carbon collected on the filter, which is blank corrected, F$^{14}$C$_m$ and M$_m$ are the measured F$^{14}$C and the measured mass of the respective carbon fraction (mvOC, OC or EC), and F$^{14}$C$_b$ and M$_b$ the fraction modern and the mass of the respective carbon fraction of blanks.

Since the directly determined system blank (i.e. the amount of $CO_2$ measured, when no sample is introduced into the aerosol combustion system) is $0.52 \pm 0.31$ µgC mvOC per extraction (i.e., M$_b$ for mvOC), it is much too small to be analyzed for F$^{14}$C$_b$ of mvOC. To do blank corrections for mvOC, we need to estimate its F$^{14}$C$_b$. We assume that F$^{14}$C$_b$ for mvOC ranges from 0 to 1 with a continuous uniform distribution as the reviewer suggested. To propagate uncertainties, a Monte Carlo simulation with 10,000 individual calculations of F$^{14}$C$_S$ for mvOC was conducted according to Eq. (S4). For calculation inputs, F$^{14}$C$_b$ of mvOC. was randomly chosen from a

continuous uniform distribution between 0 and 1. $F^{14}C_m$ for mvOC was randomly chosen from a normal distribution symmetric around the measured values with uncertainties as standard deviation (Table S4). The random values for $M_m$ and $M_b$ were taken from a triangular distribution, which has its maximum at the central value and 0 at the upper and lower limits. Then 10,000 different estimations of $F^{14}C_S$ for mvOC were calculated. The derived average represents the best estimate, and the standard deviation represents the combined uncertainties.

Figure S9 shows the $F^{14}C$ of mvOC before and after blank corrections ($F^{14}C_m$ and $F^{14}C_S$, respectively) for the contamination introduced by the isolation procedure.

As shown in Figure S9, the differences in $F^{14}C$ values of mvOC before and after blank corrections are really small, with the biggest absolute difference of 0.009 for sample Taiyuan-L. The small correction of $F^{14}C$ for mvOC will not affect any conclusion from this study. Thus, in this study, the contamination is assessed but not used for further data correction.

For OC and EC, the contamination introduced by the isolation procedure yields $0.72 \pm 0.44$ µgC EC and $0.85 \pm 0.49$ µgC OC per extraction, respectively, which are the directly determined system blank (i.e. the amount of $CO_2$ measured, when no sample is introduced into the aerosol combustion system). The blanks are less than 3% of the sample amount and thus can be neglected, compared with our sample size of 30–391 µgC EC per extraction and 30–445 µgC OC per extraction. Therefore, the blank correction will not introduce large uncertainties to the data. In this study, the contamination is assessed but not used for further data correction for $F^{14}C$ of OC and EC.

**S3S4. Gas-particle partitioning of OC fractions**

Ambient OC has a large range of volatility and contains low to high volatility compounds. The low-volatility OC is in the particle phase, and high volatility OC is mostly in the gas phase. OC with intermediate volatility are semi-volatile (SVOC) and can either be in the particle or the gas phase depending on temperature, concentrations of OC etc. (Donahue et al., 2006, 2009).

In this study, particulate OC is collected on a single bare-quartz filter. mvOC (desorbed at 200 °C in He) on the quartz filters is influenced by evaporation of SVOC from collected particles and adsorption of organic vapors to the quartz fibers. To estimate the particulate fraction of mvOC, we estimate the gas-particle partitioning of individual OC fractions using the thermal/optical OC/EC analysis and a volatility basis set (VBS) model. The VBS bins the organic aerosol (OA) compounds according to the effective saturation concentrations ($C*$). If we project concentrations

of organic aerosol (OA) onto the basis set of saturation concentrations, we can estimate the volatility distribution of the organic material. The concentrations of OA ($C_{OA}$) and the equilibrium partitioning fractions ($f_i$) at equilibrium can be described by (Donahue et al., 2006, 2011):

$$f_i = \left(1 + \frac{c_i^*}{c_{OA}}\right)^{-1} \tag{S4S5}$$

where $C_i^*$ is the $C^*$ for OC fraction $i$. If $C_i^*$ is equal to $C_{OA}$, then 50 % of OA mass will be in the particle phase, that is $f_i = 0.5$; if $C_i^* = 4 \times C_{OA}$, then the OA is more volatile, and only 20 % OA mass in the particle phase ($f_i = 0.2$).

Thermal-optical OC/EC analysis measures OC, not the OA. We use the OA/OC mass ratio to estimate OA concentrations from the OC concentrations. Then Eq. S4 S5 can be reformulated as:

$$X_{Pi} = \left(1 + \frac{c_i^*}{\frac{OA}{OC} \times c_{OC}}\right)^{-1} \tag{S5S6}$$

where $X_{Pi}$ is the particulate fraction in OC fraction $i$; $C_{OC}$ can be quantified by the thermal-optical OC/EC analysis, and is the sum of particulate OC fractions;

The $C^*$ value for each OC fraction are taken from Ma et al. (2016), where OC fractions are measured using the IMPROVE_A protocol, and are derived so that $X_{Pi}$ stands for the particulate fraction of the total carbon found on the filter for each temperature step $i$. The IMPROVE_A temperature plateaus in the pure He phase are 140 °C, 280 °C, 480 °C, 580 °C, and the corresponding thermal carbon fractions are called $OC_{140°C}$, $OC_{280°C}$, $OC_{480°C}$ and $OC_{580°C}$, where the subscript refers to the desorption temperature of the IMPROVE_A protocol, respectively. The empirical VBS consisting of bins separated by 0.5 of an order of magnitude in volatility at 300 K: the $C^*$ for $OC_{140°C}$ to $OC_{580°C}$ are $10^{1.6}$, $10^{1.1}$, $10^{0.6}$ and $10^{0.1}$ μg m$^{-3}$, respectively. The OA/OC is assumed to be 1.8 for urban environment (Turpin and Lim, 2001; Xing et al., 2013). In this study, samples in Beijing, Guangzhou, Shanghai, Taiyuan and Chongqing are measured using the IMPROVE_A protocol, thus $X_{pi}$ for samples of these five cities are estimated using the empirical $C^*$, as shown in Fig. 8a in the main text.

The particulate fraction ($X_{pi}$) of OC fractions increases considerably from $OC_{140°C}$ to $OC_{580°C}$, consistent with the temperature steps. The temperature of the OC/EC analysis is the desorption temperature rather than evaporation, but it is still reasonable to assume that the desorption temperature is closely related to the volatility. OC desorbed at higher temperature tends to have a low volatility and more likely to exist in the particle phase.

Increasing the particulate OC concentrations ($C_{OC}$ in Eq. S6) increase the particulate fraction for each OC fraction ($X_{p1}$, $X_{p2}$, $X_{p3}$ and $X_{p4}$). For example, when $C_{OC}$ increases from 8.8 μg m$^{-3}$ to 64 μg m$^{-3}$, particulate fraction of OC$_{140°C}$ increases from 0.28 (i.e., 28 % OC$_{140°C}$ in the particle phase) to 0.74 and particulate fraction of OC$_{280°C}$ increases from 0.56 to 0.90. The desorption temperature for mvOC is 200 °C, falling within the 140 °C for OC$_{140°C}$ and 280 °C for OC$_{28\theta0°C}$. We thus estimated the particulate fraction for mvOC will fall within the $X_{p1}$ for OC$_{140°C}$ and $X_{p2}$ for OC$_{28\underline{0}°C}$.

[Figure]

**Figure S1.** Geographic locations of the six Chinese sampling sites. Cities in northern China are shown as blue circle, cities in southern China as blue triangle. In northern China, large quantities of coal are used for heating during a formal residential "heating season" (15 November to 15 March next year) in winter.

[Figure]

**Figure S2.** Selected samples for $^{14}C$ analysis. 2-3 composite samples that represent high (H), medium (M) and low (L) TC concentrations are combined from several individual filter samples for each city (a. Xi'an; b. Beijing; c. Taiyuan; d. Shanghai; e. Chongqing; f. Guangzhou). Each sample is consisting of 2 to 3 24-hr filter pieces with similar TC loadings and air mass backward trajectories (Table S1).

[Figure]

**Figure S3.** Probability density function (PDF) of the relative source contributions of coal and vehicle to EC ($f_{coal}$, $f_{vehicle}$) constrained by $\delta^{13}$C, calculated using the Bayesian Markov chain Monte Carlo approach (Supplement S2). Samples collected from other cities including Xi'an, Taiyuan, Chongqing were not measured for $\delta^{13}C_{EC}$.

[Figure]

**Figure S4.** The PDF of $p$ values ($p$ is the fraction of EC from coal combustion in that from total fossil sources) for each sample, which is constrained by $\delta^{13}C_{EC}$. Calculation of $p$ is present in Sect. 2.4. Other cities including Xi'an, Taiyuan, Chongqing were not measured for $\delta^{13}C$.

[Figure]

**Figure S5.** Reproducibility of desorption performed for test filter#1, test filter#2 and winter-H. The number on the *x*-axis represent the times of mvOC desorption. **(a)** the desorbed mvOC amount in µgC per extraction; **(b)** the desorbed mvOC in µg cm$^{-2}$ calculated by dividing the desorbed mvOC amount (µgC) by the filter area (cm$^{-2}$). The horizontal black line indicates the OC1 (µg cm$^{-2}$) measured by EUSAAR_2 protocol and the shaded area denotes its uncertainties. The two red squares represent the repeated mvOC extraction for sample winter-H using larger filter pieces. The panel (a) and (b)  have the same *x*-axis.

[Figure]

**Figure S6.** An example PDF of SOC concentrations (in light blue), $SOC_{fossil}$ (light green) and $OC_{o,nf}$ (approximately $SOC_{nf}$, orange) for sample Beijing-H. The estimated SOC and $SOC_{fossil}$ concentrations are estimated by $^{14}C$-apportioned OC and EC in combination with $p$ values randomly chosen from 0–1 (Sect. 2.4). The mean and median are indicated by the dashed and solid vertical lines. The mean and median are very close to each other due to the symmetric PDF. The median (interquartile range) and mean ($\pm$ SD) values are given in Table S7 and Table S8, respectively.

[Figure]

**Figure S7. (a)** Concentrations of $mvOC_{nf}$ (green bar) and $OC_{o,nf}$ (approximately $SOC_{nf}$, orange bar); **(b)** Concentrations of $mvOC_{fossil}$ (dark green bar), and $SOC_{fossil}$. $SOC_{fossil}$ is estimated based on $^{14}C$-apportioned OC and EC, combined with primary OC/EC ratios of coal combustion and vehicle exhaust, and $p$ values. $SOC_{fossil}$ estimated using using $p$ values of 0–1 are shown in dark orange. For Chongqing, lower $p$ values of 0–0.5 are used to estimate $SOC_{fossil}$, shown as bars filled with stripes. The panel (a) and (b) has the same $x$-axis.

[Figure]

**Figure S8.** Correlation between mvOC$_{nf}$ and SOC$_{nf}$. SOC$_{nf}$ is estimated by subtracting primary OC from biomass burning (POC$_{bb}$) from OC$_{nf}$, where POC$_{bb}$ is estimated by multiplying EC$_{bb}$ with the OC/EC ratio of fresh biomass burning plumes($r_{bb}$). Here we take $r_{bb}$= 5 (3–7; mean; minimum-maximum) and $r_{bb}$= 3 (2–4), higher and lower than our best estimate of $r_{bb}$ (4, 3–5), respectively.

[Figure]

**Figure S9.** Fraction modern ($F^{14}C$) of mvOC before and after blank corrections ($F^{14}C_m$ and $F^{14}C_s$, respectively) for the contamination introduced by the isolation procedure. Detailed method to do the blank corrections is described in Supplement S3.

**Table S1.** Details of sampling information.

| | City | Sampling sites | Longitude, latitude | Sample name[*] | Sampling date | RH (%)[**] | Temperature (°C)[**] | Wind speed (m s⁻¹)[**] |
|---|---|---|---|---|---|---|---|---|
| Northern China | Xi'an (XA) | Institute of Earth Environment, Chinese Academy of Sciences | 34.2° N, 108.9° E | winter-H | 12/20/2015 12/21/2015 | 73 79 | -3~6 -1~5 | 1.0 0.6 |
| | | | | winter-M | 11/30/2015 12/8/2015 12/9/2015 | 85 85 | 3~11 2~8 2~6 | 1.1 1.1 1.0 |
| | | | | winter-L | 12/14/2015 12/16/2015 12/17/2015 | 67 50 65 | -2~8 -5~4 -4~3 | 1.9 1.3 1.2 |
| | Beijing (BJ) | Institute of Remote Sensing Applications, Chinese Academy of Sciences | 39.9° N, 116.4° E | Beijing-H | 1/15/2014 1/16/2014 | 56 67 | -4~13 -3~5 | 1.5 1.9 |
| | | | | Beijing-L | 1/1/2014 1/3/2014 | 37 50 | -2~13 -4~8 | 1.6 2.2 |
| | Taiyuan (TY) | Taiyuan University of Technology | 37.9° N, 112.5° E | Taiyuan-H | 1/14/2014 1/15/2014 1/16/2014 | 37 37 37 | -14~3 -11~4 -13~5 | 1.3 1.2 1.2 |
| | | | | Taiyuan-L | 1/12/2014 1/17/2014 1/20/2014 | 36 37 30 | -15~2 -14~3 -14~3 | 1.2 1.6 2.5 |
| Southern China | Shanghai (SH) | Fudan University | 31.2° N, 121.4° E | Shanghai-H1 | 12/30/2013 12/31/2013 | 55 49 | -3~10 -1~14 | 1.8 2.0 |
| | | | | Shanghai-H2 | 1/17/2014 1/18/2014 1/19/2014 | 70 76 67 | 5~11 -2~7 3~10 | 2.4 1.8 2.5 |
| | Chongqing (CQ) | Super Monitoring Station of Chongqing | 29.9° N, 106.5° E | Chongqing-H | 1/15/2014 1/16/2014 | 86 91 | 7~9 7~10 | 0.5 0.2 |
| | | | | Chongqing-L | 12/27/2013 1/8/2014 1/12/2014 | 92 90 89 | 6~9 6~8 4~10 | 0.4 0.3 0.5 |
| | Guangzhou (GZ) | South China Institute of Environmental Sciences, Ministry of Environment Protection | 23.1° N, 113.2° E | Guangzhou-H | 1/6/2014 1/7/2014 | 73 84 | 11~20 16~21 | 1.6 1.7 |
| | | | | Guangzhou-L | 1/9/2014 1/12/2014 | 62 67 | 11~15 8~18 | 2.8 4.9 |

[*]Composite samples that represent high (H), medium(M) and low (L) TC concentrations are combined from several individual 24-hr samples for [14]C analysis.
[**]The meteorological data is obtained from the Meteorological Institute of Shaanxi Province, Xi'an, China. Daily average relative humidity (RH), daily average wind speed (m s-1) and ambient temperature (minimum~maximum; °C). There was no precipitation in all cities during all sampling dates.

**Table S2**. Pilot tests to determine the flushing time using He before heating the filter pieces at 200 °C in He to desorb mvOC, using a test filter. The results show that the flushing time (10 min, 15 min or 60 min) does not affect the desorption amount of mvOC.

| Test | Flushing time | Desorbed amount of mvOC |
|------|---------------|-------------------------|
| 1    | 10 min        | 5.8 $\mu$g cm$^{-2}$    |
| 2    | 15 min        | 5.9 $\mu$g cm$^{-2}$    |
| 3    | 15 min        | 5.8 $\mu$g cm$^{-2}$    |
| 4    | 60 min        | 5.7 $\mu$g cm$^{-2}$    |
| 5    | 60 min        | 5.7 $\mu$g cm$^{-2}$    |

**Table S3.** Measured $F^{14}C$ values and masses of the standards with their nominal $F^{14}C$ values.

| Standards | | nominal $F^{14}C$ | measured $F^{14}C$ ($F^{14}C_m$) | measured mass ($M_m$, µgC) |
|---|---|---|---|---|
| Combustion processes[a] | OXII | 1.3406 | 1.327 ± 0.022 | 65 |
| | OXII | 1.3406 | 1.321 ± 0.012 | 117 |
| | anthracite | 0 | 0.020 ± 0.001 | 51 |
| | anthracite | 0 | 0.002 ± 0.001 | 75 |
| | anthracite | 0 | 0.008 ± 0.001 | 217 |
| | anthracite | 0 | 0.004 ± 0.001 | 219 |
| | anthracite | 0 | 0.005 ± 0.001 | 254 |

[a] For combustion processes, two sets of standard material: the oxalic acid HOxII and anthracite with known $^{14}C$ contents ($F^{14}C$ = 1.3406 and $F^{14}C$ = 0, respectively) were combusted using the aerosol combustion system (ACS) and used for quality control.

**Table S4.** Concentrations of mvOC, OC and EC (µg m$^{-3}$), the contribution of mvOC to the total OC (mvOC/OC, %), fraction modern of mvOC, OC and EC (F$^{14}$C$_{(mvOC)}$, F$^{14}$C$_{(OC)}$, F$^{14}$C$_{(EC)}$) and stable carbon isotopic composition of EC ($\delta^{13}$C$_{EC}$).

| City | Sample Name | mvOC* (µg m$^{-3}$) | OC (µg m$^{-3}$) | EC (µg m$^{-3}$) | mvOC/OC (%) | F$^{14}$C$_{(mvOC)}$ | F$^{14}$C$_{(OC)}$ | F$^{14}$C$_{(EC)}$ | $\delta^{13}$C$_{EC}$ (‰) |
|---|---|---|---|---|---|---|---|---|---|
| Xi'an (XA) | winter-H | 6.5 | 47.1 | 9.9 | 13.9 | 0.517 ± 0.006 | 0.640 ± 0.009 | 0.340 ± 0.005 | |
| | winter-M | 3.6 | 37.9 | 6.1 | 9.5 | 0.529 ± 0.007 | 0.609 ± 0.007 | 0.258 ± 0.005 | |
| | winter-L | 1.6 | 14.5 | 2.8 | 11.2 | 0.483 ± 0.050** | 0.626 ± 0.007 | 0.320 ± 0.005 | |
| Beijing (BJ) | Beijing-H | 7.4 | 50.4 | 6.9 | 14.6 | 0.290 ± 0.008 | 0.366 ± 0.003 | 0.253 ± 0.002 | -23.96 ± 0.02 |
| | Beijing-L | 0.7 | 21.8 | 4.7 | 3.3 | 0.334 ± 0.008 | 0.413 ± 0.004 | 0.266 ± 0.003 | -24.06 ± 0.02 |
| Taiyuan (TY) | Taiyuan-H | 2.4 | 45.1 | 14.8 | 5.3 | NA*** | 0.320 ± 0.030 | 0.106 ± 0.002 | |
| | Taiyuan-L | 2.9 | 23.2 | 8.2 | 12.7 | 0.241 ± 0.008 | 0.318 ± 0.023 | 0.105 ± 0.003 | |
| Shanghai (SH) | Shanghai-H1 | 2.7 | 24.7 | 5.2 | 10.9 | 0.392 ± 0.008 | 0.609 ± 0.004 | 0.332 ± 0.003 | -25.45 ± 0.03 |
| | Shanghai-H2 | 1.7 | 18.2 | 4.2 | 9.3 | 0.443 ± 0.008 | 0.580 ± 0.004 | 0.303 ± 0.003 | -24.72 ± 0.02 |
| Chongqing (CQ) | Chongqing-H | 6.7 | 46.7 | 11.8 | 14.3 | 0.595 ± 0.011 | 0.839 ± 0.029 | 0.414 ± 0.010 | |
| | Chongqing-L | 3 | 29 | 8.1 | 10.2 | 0.562 ± 0.010 | 0.833 ± 0.028 | 0.466 ± 0.026 | |
| Guangzhou (GZ) | Guangzhou-H | 2.4 | 23.4 | 4.5 | 10.3 | 0.481 ± 0.008 | 0.592 ± 0.004 | 0.271 ± 0.002 | -26.01 ± 0.02 |
| | Guangzhou-L | 1 | 8.8 | 2.5 | 10.9 | NA*** | 0.667 ± 0.005 | 0.375 ± 0.004 | -26.15 ± 0.02 |

* Excellent agreement was found between mvOC amount measured using ACS system and OC1 following EUSAAR_2 using a Sunset carbon analyzer, except for sample winter-L. Here we took the OC1 by EUSAAR_2 as mvOC concentrations (both desorbed at 200 °C in He) for samples at Xi'an. Blank corrections were performed for OC1 mass, the mvOC concentrations in Xi'an were therefore blank corrected. ** For sample winter-L, we take the measured F$^{14}$C$_{(mvOC)}$ values but assign a bigger absolute uncertainty of 0.05, due to their low mvOC recoveries. *** Samples get lost during the $^{14}$C measurements.

**Table S5.** Relative non-fossil sources contribution to mvOC, OC, EC and mrOC ($f_{nf}$(mvOC), $f_{nf}$(OC), $f_{bb}$(EC), $f_{nf}$(mrOC)), and relative fossil sources contribution to mvOC, OC, EC and mrOC($f_{fossil}$(mvOC), $f_{fossil}$(OC), $f_{fossil}$(EC), $f_{fossil}$(mrOC)).

| City | Sample Name | $f_{nf}$(mvOC) | $f_{nf}$(OC) | $f_{bb}$(EC) | $f_{nf}$(mrOC)[*] | $f_{fossil}$(mvOC) | $f_{fossil}$(OC) | $f_{fossil}$(EC) | $f_{fossil}$(mrOC)[*] |
|---|---|---|---|---|---|---|---|---|---|
| Xi'an | winter-H | 0.475 ± 0.011 | 0.587 ± 0.014 | 0.309 ± 0.008 | 0.606 ± 0.015 | 0.525 ± 0.011 | 0.413 ± 0.014 | 0.691 ± 0.008 | 0.394 ± 0.015 |
| (XA) | winter-M | 0.485 ± 0.011 | 0.558 ± 0.012 | 0.235 ± 0.006 | 0.566 ± 0.013 | 0.515 ± 0.011 | 0.442 ± 0.012 | 0.765 ± 0.006 | 0.434 ± 0.013 |
|  | winter-L | 0.443 ± 0.047 | 0.574 ± 0.012 | 0.291 ± 0.007 | 0.591 ± 0.014 | 0.557 ± 0.047 | 0.426 ± 0.012 | 0.709 ± 0.007 | 0.409 ± 0.014 |
| Beijing | Beijing-H | 0.266 ± 0.009 | 0.336 ± 0.007 | 0.230 ± 0.005 | 0.348 ± 0.008 | 0.734 ± 0.009 | 0.664 ± 0.007 | 0.770 ± 0.005 | 0.652 ± 0.008 |
| (BJ) | Beijing-L | 0.306 ± 0.009 | 0.379 ± 0.008 | 0.242 ± 0.005 | 0.382 ± 0.008 | 0.694 ± 0.009 | 0.621 ± 0.008 | 0.758 ± 0.005 | 0.618 ± 0.008 |
| Taiyuan | Taiyuan-H | NA[**] | 0.293 ± 0.028 | 0.096 ± 0.003 | NA | NA | 0.707 ± 0.028 | 0.904 ± 0.003 | NA |
| (TY) | Taiyuan-L | 0.221 ± 0.009 | 0.292 ± 0.022 | 0.095 ± 0.003 | 0.302 ± 0.025 | 0.779 ± 0.009 | 0.708 ± 0.022 | 0.905 ± 0.003 | 0.698 ± 0.025 |
| Shanghai | Shanghai-H1 | 0.360 ± 0.010 | 0.559 ± 0.011 | 0.302 ± 0.006 | 0.584 ± 0.012 | 0.640 ± 0.010 | 0.441 ± 0.011 | 0.698 ± 0.006 | 0.416 ± 0.012 |
| (SH) | Shanghai-H2 | 0.406 ± 0.011 | 0.532 ± 0.011 | 0.275 ± 0.006 | 0.545 ± 0.011 | 0.594 ± 0.011 | 0.468 ± 0.011 | 0.725 ± 0.006 | 0.455 ± 0.011 |
| Chongqing | Chongqing-H | 0.546 ± 0.014 | 0.771 ± 0.030 | 0.377 ± 0.011 | 0.809 ± 0.035 | 0.454 ± 0.014 | 0.229 ± 0.030 | 0.623 ± 0.011 | 0.191 ± 0.035 |
| (CQ) | Chongqing-L | 0.516 ± 0.014 | 0.764 ± 0.030 | 0.423 ± 0.025 | 0.794 ± 0.033 | 0.484 ± 0.014 | 0.236 ± 0.030 | 0.577 ± 0.025 | 0.206 ± 0.033 |
| Guangzhou | Guangzhou-H | 0.441 ± 0.011 | 0.543 ± 0.011 | 0.246 ± 0.005 | 0.555 ± 0.011 | 0.559 ± 0.011 | 0.457 ± 0.011 | 0.754 ± 0.005 | 0.445 ± 0.011 |
| (GZ) | Guangzhou-L | NA[**] | 0.612 ± 0.012 | 0.341 ± 0.007 | NA | NA | 0.388 ± 0.012 | 0.659 ± 0.007 | NA |

[*] $f_{nf}$(mrOC) is calculated by the differences between OC and mvOC (Sect. 2.4); [**] Samples get lost during the $^{14}$C measurements.

**Table S6.** Concentrations of mvOC, OC and EC from non-fossil sources (mvOC$_{nf}$, OC$_{nf}$ and EC$_{bb}$) and fossil sources (mvOC$_{fossil}$, OC$_{fossil}$ and EC$_{fossil}$) in units of μg m$^{-3}$.

| City | Sample Name | mvOC$_{nf}$ | mvOC$_{fossil}$ | OC$_{nf}$ | OC$_{fossil}$ | EC$_{bb}$ | EC$_{fossil}$ |
|---|---|---|---|---|---|---|---|
| Xi'an | winter-H | 3.11 ± 0.19 | 3.44 ± 0.21 | 27.69 ± 1.53 | 19.46 ± 1.17 | 3.08 ± 0.18 | 6.86 ± 0.37 |
| (XA) | winter-M | 1.74 ± 0.11 | 1.85 ± 0.12 | 21.15 ± 1.17 | 16.73 ± 0.97 | 1.44 ± 0.09 | 4.70 ± 0.26 |
| | winter-L | 0.72 ± 0.09 | 0.91 ± 0.10 | 8.32 ± 0.47 | 6.17 ± 0.37 | 0.82 ± 0.05 | 1.99 ± 0.12 |
| Beijing | Beijing-H | 1.96 ± 0.12 | 5.41 ± 0.28 | 16.91 ± 1.73 | 33.47 ± 3.37 | 1.58 ± 0.32 | 5.29 ± 1.06 |
| (BJ) | Beijing-L | 0.22 ± 0.01 | 0.49 ± 0.03 | 8.25 ± 0.84 | 13.49 ± 1.36 | 1.13 ± 0.23 | 3.55 ± 0.73 |
| Taiyuan | Taiyuan-H | NA | NA | 13.22 ± 1.85 | 31.86 ± 3.42 | 1.42 ± 0.29 | 13.38 ± 2.69 |
| (TY) | Taiyuan-L | 0.65 ± 0.04 | 2.29 ± 0.12 | 6.76 ± 0.84 | 16.42 ± 1.72 | 0.78 ± 0.16 | 7.40 ± 1.48 |
| Shanghai | Shanghai-H1 | 0.97 ± 0.06 | 1.72 ± 0.09 | 13.79 ± 1.40 | 10.88 ± 1.13 | 1.58 ± 0.32 | 3.64 ± 0.73 |
| (SH) | Shanghai-H2 | 0.69 ± 0.04 | 1.01 ± 0.05 | 9.71 ± 0.99 | 8.54 ± 0.87 | 1.16 ± 0.23 | 3.05 ± 0.61 |
| Chongqing | Chongqing-H | 3.66 ± 0.21 | 3.04 ± 0.18 | 35.97 ± 3.89 | 10.73 ± 1.76 | 4.44 ± 0.90 | 7.34 ± 1.47 |
| (CQ) | Chongqing-L | 1.53 ± 0.09 | 1.43 ± 0.08 | 22.14 ± 2.39 | 6.81 ± 1.11 | 3.39 ± 0.70 | 4.62 ± 0.94 |
| Guangzhou | Guangzhou-H | 1.06 ± 0.06 | 1.34 ± 0.07 | 12.70 ± 1.29 | 10.66 ± 1.09 | 1.11 ± 0.22 | 3.40 ± 0.68 |
| (GZ) | Guangzhou-L | NA | NA | 5.39 ± 0.54 | 3.43 ± 0.36 | 0.84 ± 0.17 | 1.62 ± 0.32 |

**Table S7.** Concentrations of SOC, SOC from fossil sources ($SOC_{fossil}$) and OC from non-fossil sources excluding primary biomass burning ($OC_{o,nf}$) and fraction of fossil in SOC ($f_{fossil}(SOC)$) (median and interquartile range).

| City | Sample Name | $f_{fossil}(SOC)$ | | | SOC concentrations (µg m$^{-3}$) | | |
|---|---|---|---|---|---|---|---|
| | | $p$ (0–1)[a] | $^{13}$C-constrained $p$[b] | $p$ (0–0.5)[c] | $p$ (0–1) | $^{13}$C-constrained $p$ | $p$ (0–0.5) |
| Xi'an (XA) | winter-H | 0.37 (0.29–0.42) | | | 24.25 (21.41–27.10) | | |
| | winter-M | 0.38 (0.34–0.42) | | | 24.83 (22.79–26.84) | | |
| | winter-L | 0.38 (0.32–0.43) | | | 8.15 (7.28–8.99) | | |
| Beijing (BJ) | Beijing-H | 0.70 (0.68–0.72) | 0.70 (0.68–0.71) | | 36.01 (31.67–40.13) | 35.20 (32.75–37.82) | |
| | Beijing-L | 0.68 (0.64–0.72) | 0.67 (0.63–0.70) | | 11.80 (9.48–14.06) | 11.49 (9.99–12.78) | |
| Taiyuan (TY) | Taiyuan-H | 0.61 (0.47–0.69) | | | 19.13 (12.56–24.90) | | |
| | Taiyuan-L | 0.61 (0.44–0.69) | | | 8.88 (5.30–12.04) | | |
| Shanghai (SH) | Shanghai-H1 | 0.41 (0.35–0.47) | 0.43 (0.38–0.46) | | 12.87 (10.22–15.36) | 12.85 (11.28–14.38) | |
| | Shanghai-H2 | 0.43 (0.35–0.49) | 0.42 (0.36–0.47) | | 8.96 (6.87–10.87) | 8.74 (7.44–9.90) | |
| Chongqing (CQ) | Chongqing-H | -0.01 (-0.23–0.12) | | 0.10 (0–0.17) | 17.70 (12.03–23.06) | | 20.20 (15.05–25.20) |
| | Chongqing-L | 0.04 (-0.25–0.19) | | 0.14 (0.02–0.22) | 8.34 (4.50–12.06) | | 9.84 (6.25–13.31) |
| Guangzhou (GZ) | Guangzhou-H | 0.40 (0.34–0.45) | 0.43 (0.39–0.46) | | 13.69 (11.37–15.96) | 14.30 (12.93–15.50) | |
| | Guangzhou-L | 0.34 (0.18–0.44) | 0.36 (0.25–0.44) | | 2.99 (1.84–4.06) | 3.15 (2.49–3.79) | |

| City | Sample Name | SOC$_{fossil}$ | | | OC$_{o,nf}$ (approximately SOC$_{nf}$) |
|---|---|---|---|---|---|
| | | $p$ (0–1) | $^{13}$C-constrained $p$ | $p$ (0–0.5) | |
| Xi'an (XA) | winter-H | 8.86 (6.49–11.28) | | | 15.42 (13.98–16.82) |
| | winter-M | 9.46 (7.77–11.13) | | | 15.38 (14.46–16.31) |
| | winter-L | 3.10 (2.37–3.80) | | | 5.05 (4.64–5.48) |
| Beijing (BJ) | Beijing-H | 25.41 (22.33–28.29) | 24.55 (22.69–26.58) | | 10.60 (9.13–12.15) |
| | Beijing-L | 8.09 (6.45–9.66) | 7.63 (6.60–8.82) | | 3.74 (2.82–4.63) |
| Taiyuan (TY) | Taiyuan-H | 11.63 (5.74–16.78) | | | 7.49 (6.02–9.01) |
| | Taiyuan-L | 5.27 (2.05–8.07) | | | 3.64 (2.90–4.39) |
| Shanghai (SH) | Shanghai-H1 | 5.36 (3.78–6.85) | 5.39 (4.23–6.45) | | 7.54 (6.19–8.88) |
| | Shanghai-H2 | 3.92 (2.60–5.11) | 3.62 (2.72–4.55) | | 5.07 (4.12–6.05) |
| Chongqing (CQ) | Chongqing-H | -0.37 (-3.58–2.38) | | 2.00 (-0.10–3.91) | 18.24 (14.51–21.93) |
| | Chongqing-L | -0.16 (-2.14–1.58) | | 1.27 (0.03–2.52) | 8.65 (5.94–11.27) |
| Guangzhou (GZ) | Guangzhou-H | 5.52 (3.99–6.87) | 6.12 (5.10–6.95) | | 8.26 (7.16–9.37) |
| | Guangzhou-L | 0.99 (0.28–1.57) | 1.19 (0.68–1.56) | | 2.06 (1.41–2.67) |

[a] $p$ is the contribution of coal combustion to fossil EC (Eq. 12), $p$ was randomly chosen from 0–1, that is no constrains on $p$ values.
[b] Samples taken from Beijing, Shanghai and Guangzhou were measured for both F$^{14}$C$_{(EC)}$ and $\delta^{13}$C$_{EC}$, and $p$ in these three cities is further constrained by $\delta^{13}$C$_{EC}$ (Supplement S2).
[c] For Chongqing, we did a sensitive analysis for $p$ and tried a smaller $p$ of 0–0.5.

**Table S8.** Concentrations of SOC, SOC from fossil sources ($SOC_{fossil}$) and non-fossil sources ($OC_{o,nf}$) and fraction of fossil in SOC ($f_{fossil}(SOC)$) ($\mu g\ m^{-3}$; mean ± standard deviation).

| City | Sample Name | SOC concentrations | | $SOC_{fossil}$ | | $OC_{o,nf}$ (approximately $SOC_{nf}$) |
|---|---|---|---|---|---|---|
| | | $p\ (0–1)^a$ | $p\ (0–0.5)^b$ | $p\ (0–1)$ | $p\ (0–0.5)$ | |
| Xi'an (XA) | winter-H | 24.17 ± 4.12 | | 8.79 ± 3.17 | | 15.38 ± 2.13 |
| | winter-M | 24.79 ± 2.91 | | 9.41 ± 2.25 | | 15.38 ± 1.36 |
| | winter-L | 8.13 ± 1.24 | | 3.08 ± 0.94 | | 5.05 ± 0.61 |
| Beijing (BJ) | Beijing-H | 35.87 ± 6.29 | | 25.26 ± 4.40 | | 10.61 ± 2.24 |
| | Beijing-L | 11.67 ± 3.36 | | 7.96 ± 2.33 | | 3.71 ± 1.32 |
| Taiyuan (TY) | Taiyuan-H | 18.56 ± 9.12 | | 11.02 ± 7.96 | | 7.54 ± 2.26 |
| | Taiyuan-L | 8.60 ± 4.90 | | 4.94 ± 4.27 | | 3.65 ± 1.11 |
| Shanghai (SH) | Shanghai-H1 | 12.75 ± 3.80 | | 5.24 ± 2.22 | | 7.51 ± 1.98 |
| | Shanghai-H2 | 8.84 ± 2.97 | | 3.78 ± 1.85 | | 5.05 ± 1.44 |
| Chongqing (CQ) | Chongqing-H | 17.40 ± 8.33 | 20.0 ± 7.6 | -0.72 ± 4.28 | 1.80 ± 3.04 | 18.12 ± 5.59 |
| | Chongqing-L | 8.13 ± 5.65 | 9.8 ± 5.1 | -0.38 ± 2.71 | 1.23± 1.90 | 8.51 ± 3.97 |
| Guangzhou (GZ) | Guangzhou-H | 13.62 ± 3.39 | | 5.38 ± 2.10 | | 8.25 ± 1.64 |
| | Guangzhou-L | 2.95 ± 1.67 | | 0.90 ± 0.93 | | 2.04 ± 0.94 |

[a] $p$ is the contribution of coal combustion to fossil EC (Eq. 12), $p$ was randomly chosen from 0–1, that is no constrains on $p$ values.
[b] For Chongqing, we did a sensitive analysis for $p$ and tried a smaller $p$ of 0–0.5.